# Dissecting super-enhancer hierarchy based on chromatin interactions

Jialiang Huang[1,2], Kailong Li[3], Wenqing Cai[2], Xin Liu[3], Yuannyu Zhang[3], Stuart H. Orkin[2,4], Jian Xu[3] & Guo-Cheng Yuan [1]

Recent studies have highlighted super-enhancers (SEs) as important regulatory elements for gene expression, but their intrinsic properties remain incompletely characterized. Through an integrative analysis of Hi-C and ChIP-seq data, here we find that a significant fraction of SEs are hierarchically organized, containing both hub and non-hub enhancers. Hub enhancers share similar histone marks with non-hub enhancers, but are distinctly associated with cohesin and CTCF binding sites and disease-associated genetic variants. Genetic ablation of hub enhancers results in profound defects in gene activation and local chromatin landscape. As such, hub enhancers are the major constituents responsible for SE functional and structural organization.

[1] Department of Biostatistics and Computational Biology, Dana-Farber Cancer Institute and Harvard T.H. Chan School of Public Health, Boston, MA 02215, USA. [2] Division of Hematology/Oncology, Boston Childrens Hospital and Department of Pediatric Oncology, Dana-Farber Cancer Institute, Harvard Medical School, Boston, MA 02215, USA. [3] Department of Pediatrics, Childrens Medical Center Research Institute, University of Texas Southwestern Medical Center, Dallas, TX 75390, USA. [4] Howard Hughes Medical Institute, Boston, MA 02215, USA. These authors contributed equally: Jialiang Huang, Kailong Li. Correspondence and requests for materials should be addressed to J.X. (email: jian.xu@utsouthwestern.edu) or to G.-C.Y. (email: gcyuan@jimmy.harvard.edu)

Enhancers are *cis*-acting DNA sequences that control cell-type specific gene expression[1]. Super-enhancers (SEs) are putative enhancer clusters with unusually high levels of enhancer activity and enrichment of enhancer-associated chromatin features including occupancy of master regulators, coactivators, Mediators and chromatin factors[2–4]. SEs are often in close proximity to critical cell identity-associated genes, supporting a model in which a small set of lineage-defining SEs determine cell identity in development and disease.

Despite the proposed prominent roles, the structural and functional differences between SEs and regular enhancers (REs) remain poorly understood[5]. A few SEs have been dissected by genetic manipulation of individual constituent enhancers. In some studies, the results are consistent with a model whereby SEs are composed of a hierarchy of both essential and dispensable constituent enhancers to coordinate gene transcription[6–9]. However, due to the technical challenges in systematic characterization of SEs on a larger scale, it remains unknown the generality of hierarchical SE organization in the mammalian genome.

Enhancer activities are mediated by the 3D chromatin interactions. Recent advances in Hi-C[10] and ChIA-PET[11] technologies have enabled systematic interrogation of the genome-wide landscapes of chromatin interactions across multiple cell types and growth conditions[12–19]. These data strongly indicate that the 3D chromatin organization is highly modular, containing compartments, topologically associating domains (TADs), and insulated neighborhoods. Of note, genomic loci with high frequency of chromatin interactions are highly enriched for SEs[20–23], suggesting that proper 3D chromatin configuration may be essential for orchestrating SE activities.

Here we develop an approach to dissect the compositional organization of SEs based on long-range chromatin interactions. We find that a subset of SEs exhibits a hierarchical structure, and hub enhancers within hierarchical SEs play distinct roles in chromatin organization and gene activation. Our findings also identify a critical role for CTCF in organizing the structural (and hence functional) hierarchy of SEs.

## Results

**A subset of SEs contains hierarchical structure.** To systematically characterize the structural organization of SEs, we developed a computational approach that integrates high resolution Hi-C and ChIP-seq data (Fig. 1a). We defined SEs using the standard ROSE algorithms[2]. Briefly, neighboring enhancer elements defined based on H3K27ac ChIP-seq peaks were merged and ranked based on the H3K27ac ChIP-seq signal, and top ranked regions were designated as SEs. To quantify the degree of structural hierarchy associated with each SE, we defined a computational metric, called hierarchical score (or $H$-score for short), as follows. First, we divided each SE into 5 kb bins to match the resolution of Hi-C data (Fig. 1b). Next, we standardized the frequency of chromatin interactions for the bins of each SE to z-scores. Third, we evaluated the maximum z-score across all bins in each SE, and referred to the outcome as the $H$-score associated with the SE. A higher $H$-score value indicates the chromatin interactions associated with a SE are mediated through a small subset of constitutive elements (Fig. 1b). Fourth, by applying a threshold value of $H$-score, we divided all SEs into two categories, to which we referred as hierarchical and non-hierarchical SEs, respectively (Fig. 1b). Finally, if an enhancer element within hierarchical SEs overlaps with a bin associated with a z-score greater than the threshold $H$-score, the element is referred to a hub enhancer, whereas the remaining enhancers at the same SE are termed non-hub enhancers (Fig. 1b).

We applied this pipeline to dissect SE hierarchy in two human cell lines K562 (erythroleukemia cells) and GM12878 (B-lymphoblastoid cells), using publicly available high-resolution Hi-C and ChIP-seq data[15,24]. In total, we identified 843 and 834 SEs in K562 and GM12878 cells, respectively. By comparing high-

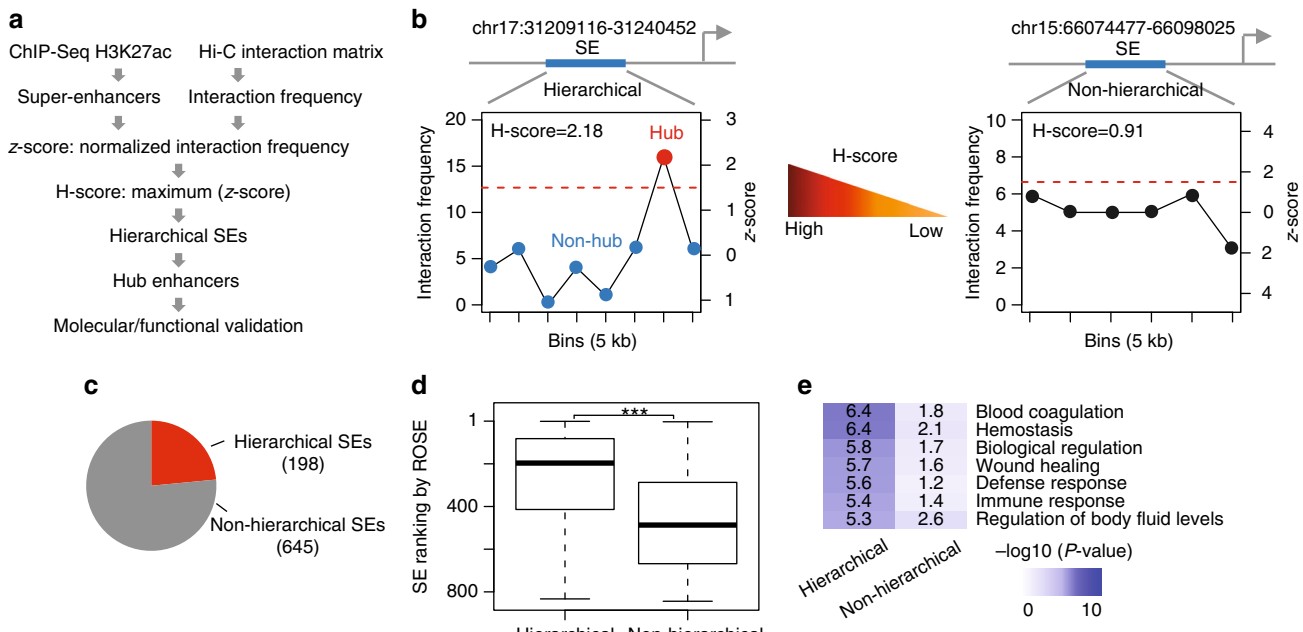

**Fig. 1** Definition of hierarchical SEs and hub enhancers based on Hi-C chromatin interactions in K562 cells. **a** Overview of pipeline. **b** Representative hierarchical (left) and non-hierarchical (right) SEs. For each 5 kb bin within SE, the frequency of chromatin interactions (left *y*-axis) of and the *z*-score (right *y*-axis) is shown. The dashed red line represents the threshold of *z*-score = 1.5. **c** The proportion of hierarchical and non-hierarchical SEs. (**d**) The ROSE ranking of hierarchical and non-hierarchical SEs. In box plots, the center line represents the median, the box limits represent the 25th and 75th percentiles and the whiskers represent the 5th and 95th percentiles. *P* values were calculated using Wilcoxon rank-sum test. *$P < 0.05$; **$P < 0.01$; ***$P < 0.001$. (**e**) GREAT functional analysis of hierarchical and non-hierarchical SEs

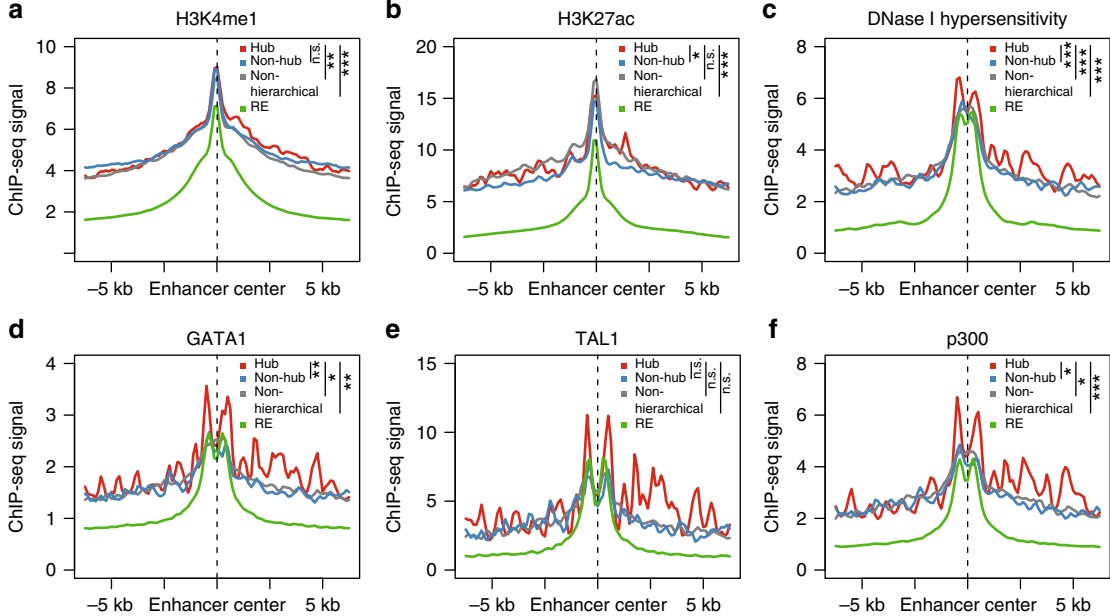

**Fig. 2** Chromatin landscapes at hub enhancers in K562 cells. **a–f** Spatial distribution of chromatin marks centered by enhancers in four groups, hub (n = 444), non-hub (n = 2303) enhancers, enhancers in non-hierarchical SEs (n = 4290) and regular enhancers (n = 22477): H3K4me1 (**a**), H3K27ac (**b**), DNase I hypersensitivity (**c**), master regulators GATA1 (**d**) and TAL1 (**e**), coactivator p300 (**f**). P values were calculated using Student's t-test based on the ChIP-seq signal intensity within 1 kb window centered by enhancers. *P < 0.05; **P < 0.01; ***P < 0.001, n.s. not significant

resolution (5 kb) Hi-C profiles with annotated enhancers in K562 and GM12878 cells[13], we observed that SEs contain a significantly higher frequency of chromatin interactions than REs (P = 1.2E −69 in K562, P = 2.0E−123 in GM12878, Student's t-test, Supplementary Fig. 1a), consistent with previous studies[20,21]. By applying a threshold value of H-score = 1.5, which roughly corresponds to the 95th percentile of z-scores (Supplementary Fig. 1b), we divided SEs into two categories: hierarchical and non-hierarchical SEs (Supplementary Fig. 1c). We observed hub enhancers tend to be broader than non-hub enhancers (Supplementary Fig. 1d). We also found the properties of hub enhancers are not sensitive to the specific choice of H-score threshold, as described in the following sections.

In total, we identified 198 (23% of all SEs) and 286 hierarchical SEs (34%) in K562 and GM12878 cells, respectively (Fig. 1c and Supplementary Fig. 2a). The hierarchical SEs tend to rank higher than non-hierarchical SEs based on the ROSE algorithm (P = 1.2E−25 in K562, P = 2.5E−21 in GM12878, Wilcoxon rank-sum test, respectively, Fig. 1d and Supplementary Fig. 2b). By GREAT functional analysis[25], we observed that, compared with non-hierarchical SEs, hierarchical SEs were more enriched with gene ontology (GO) terms associated with cell-type-specific biological processes, such as 'blood coagulation' in K562 cells and 'B cell homeostasis' in GM12878 cells (Fig. 1e and Supplementary Fig. 2c). These results suggest that hierarchical SEs may play a more important role in the maintenance of cell identity.

**Hub and non-hub enhancers share similar chromatin landscapes.** To further investigate the molecular differences between hub and non-hub enhancers within hierarchical SEs, we compared the spatial patterns of histone marks among three enhancer groups: hub, non-hub and REs. Compared with non-hub enhancers, hub enhancers display no significant difference in H3K4me1 ChIP-seq signal (Fig. 2a and Supplementary Fig. 3a). The signals for H3K27ac and DNase I hypersensitivity are slightly higher at hub than other types of enhancers (Fig. 2b, c and Supplementary Fig. 3b, c); however, the difference is subtle and

we cannot exclude the possibility that it may be caused by experimental variation.

One of the hallmark features of SEs is the enrichment of cell-type-specific master regulators and coactivators[2]. We then compared the distribution of transcription factor (TF) binding profiles. Hub enhancers contain significantly higher ChIP-seq binding signals for lineage-regulating master regulators than non-hub enhancers, such as GATA1 and TAL1 in K562 cells, and PAX5 and EBF1 in GM12878 cells (Fig. 2d, e and Supplementary Fig. 3d, e), although the differences are moderate. Hub enhancers also display increased occupancy of histone acetyltransferase p300, a coactivator associated with active enhancers (Fig. 2f and Supplementary Fig. 3f). Overall, the TF binding profiles at enhancers within non-hierarchical SEs and non-hub enhancers are highly similar (Fig. 2 and Supplementary Fig. 3). Taken together, these results demonstrate that hub and non-hub enhancers are characterized by moderate differences in the occupancy of active enhancer-associated histone modifications and lineage-specifying TFs.

**Hub enhancers are enriched with cohesin and CTCF binding.** Since hub and non-hub enhancers are defined based on the frequency of chromatin interactions, we next compared the occupancy of cohesin and CTCF, two factors essential for mediating long-range enhancer–promoter interactions and DNA looping[26]. To this end, we compared the enhancer groups with the ChIP-seq profiles for CTCF and two cohesin components, SMC3 and RAD21. Compared with non-hub enhancers, the occupancy of all three factors is markedly elevated at hub enhancers (Fig. 3a–c and Supplementary Fig. 4a-c), consistent with a critical role of CTCF and cohesin in mediating chromatin interactions associated with hub enhancers. Importantly, while the role of CTCF in mediating chromatin organization, such as TADs, has been well established[14], its association with SE constituents has not been previously reported. In fact, only a small fraction (6% in K562; 24% in GM12878) of hub enhancers overlap with known TAD boundaries (Fig. 3d and Supplementary Fig. 4d), which is

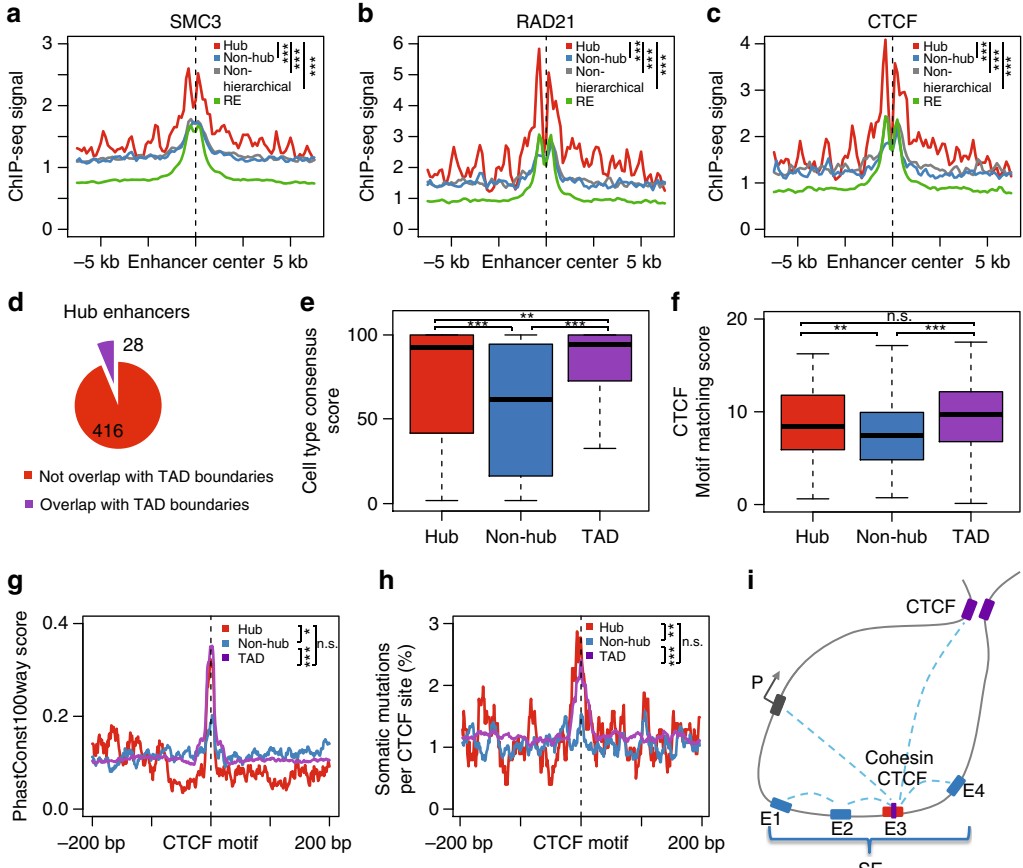

**Fig. 3** CTCF binding at hub enhancers within hierarchical SEs in K562 cells. **a**–**c** Spatial distribution of two cohesin components SMC3 (**a**) and RAD21(**b**), and CTCF (**c**), centered by enhancers in four groups. *P* values were calculated using Student's *t*-test based on the ChIP-seq signal intensity of 1 kb window centered by enhancers. *$P < 0.05$; **$P < 0.01$; ***$P < 0.001$, n.s. not significant. **d** Percentage of hub enhancers with (purple) or without (red) overlapping with TAD boundaries collected from the literature[15]. The CTCF ChIP-seq peaks/motif-sites associated with hub enhancers overlapping with TAD boundaries were excluded for analysis in **e**–**h**. **e,f** CTCF binding consensus across cell types (**e**) and CTCF-motif-matching score (**f**) of CTCF peaks in different contexts: hub (red), non-hub enhancers (blue) and TAD boundaries (purple). For each CTCF peak in K562, the consensus score (y-axis) was quantified as the percentage of cell types containing the same CTCF peak. In box plots, the center line represents the median, the box limits represent the 25th and 75th percentiles and the whiskers represent the 5th and 95th percentiles. *P*-values were calculated using Student's *t*-test. *$P < 0.05$; **$P < 0.01$; ***$P < 0.001$, n.s. not significant. **g** Sequence conservation around CTCF motif sites. The sitepro plots were centered by CTCF motif sites. *P* values were calculated using Student's *t*-test based on the PhastConst100way score (y-axis) within CTCF motif sites. *$P < 0.05$; **$P < 0.01$; ***$P < 0.001$, n.s. not significant. **h** Somatic mutation rate in cancers collected from IGGC around CTCF motif sites. The sitepro plots were centered by CTCF motif sites with 10 bp smoothing window. *P*-values were calculated using Fisher's exact test based on overlap between CTCF motif sites and somatic mutation sites. *$P < 0.05$; **$P < 0.01$; ***$P < 0.001$, n.s. not significant. **i** Model of the hierarchical organization of SEs containing both hub and non-hub enhancers. A hub enhancer is highly enriched with CTCF and cohesin binding, and functions as an organization hub to coordinate the non-hub enhancers and other distal regulatory elements within and beyond the SE

comparable to the genome-wide frequency of CTCF peaks overlapping with TAD boundaries, highlighting a TAD-independent role of CTCF.

To identify potential contextual differences between CTCF binding associated with distinct functions, we divided the CTCF ChIP-seq peaks into three non-overlapping subsets that overlap with hub enhancers, non-hub enhancers or TAD boundaries, respectively. To further distinguish CTCF binding at distinct regulatory regions, we excluded peaks overlapping with both hub enhancers and TAD boundaries (Fig. 3d and Supplementary Fig. 4d). We first examined the cross cell-type variability of CTCF binding based on CTCF ChIP-seq signals in 55 cell types from ENCODE[24]. Consistent with previous studies[14,27], we found that CTCF binding sites associated with TAD boundaries are highly conserved (Fig. 3e and Supplementary Fig. 4e). In addition, within SEs, CTCF sites associated with hub enhancers are more conserved than those associated with non-hub enhancers. We

hypothesized that the cell-type variability of CTCF binding may reflect the binding affinity of CTCF to its cognate sequences, which can be quantified by the motif-matching scores. Therefore, we compared the distribution of motif scores associated with different subsets of CTCF binding sites. The motif scores for CTCF sites associated with TAD boundaries and hub enhancers are higher than non-hub enhancer-associated CTCF sites, consistent with the CTCF ChIP-seq signal intensity (Fig. 3f and Supplementary Fig. 4f). Of note, a similar pattern is observed for the genomic sequence conservation of CTCF binding sites as quantified by the phastCons100way score (Fig. 3g and Supplementary Fig. 4g), suggesting that the cell-type variation associated with CTCF binding may be under evolutionary pressure.

Somatic mutations of TAD or insulated neighborhood boundaries have been reported in cancer[28–30]. Consistently, we observed high frequency of somatic mutations in TAD boundary-associated CTCF sites using somatic mutations in human cancers

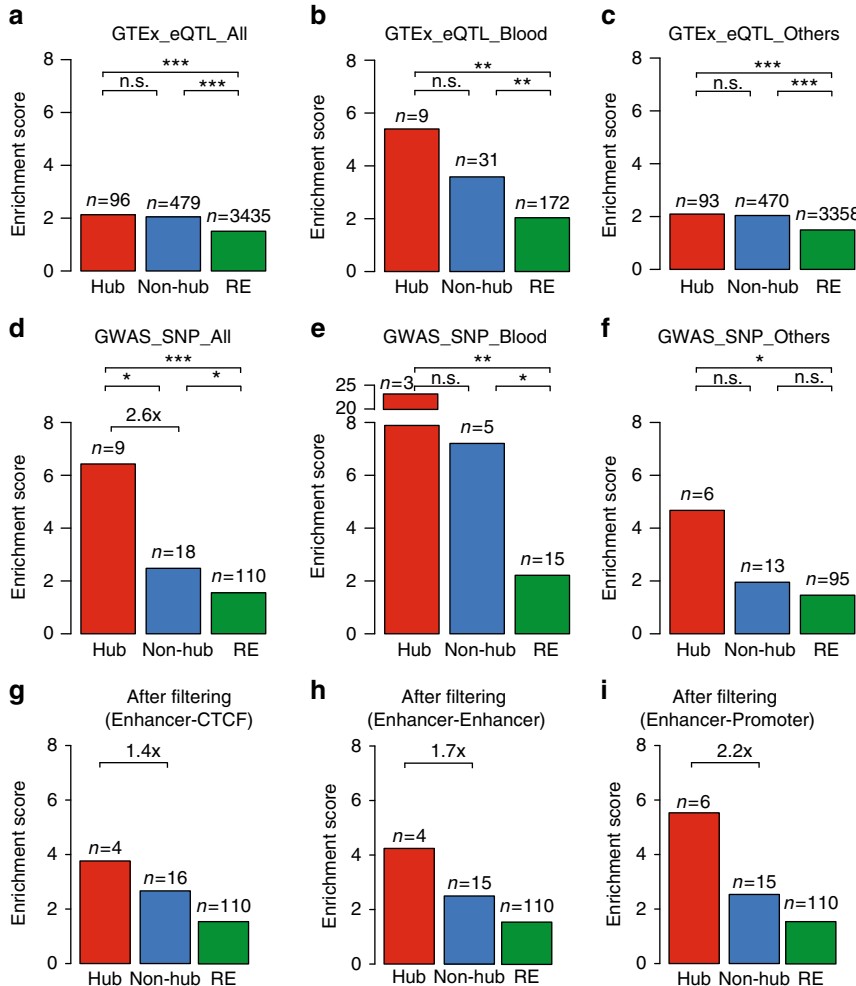

**Fig. 4** Enrichment of genetic variants associated with cell-type-specific gene expression and diseases in hub enhancers in K562 cells. **a–c** Enrichment of the eQTLs curated in GTEx in the enhancers in three groups, using randomly selected genomic regions as control (see Methods). The GTEx eQTL identified in all tissues (**a**) were separated into two subsets, identified in whole blood (**b**) or other tissues (**c**). The number of enhancers overlap with eQTLs in each group was labelled on each bar. $P$ values were calculated using Fisher's exact test. *$P < 0.05$; **$P < 0.01$; ***$P < 0.001$, n.s. not significant. **d–f** Enrichment of the disease or traits-associated SNPs curated in GWAS catalog in the enhancers in three groups, using randomly selected genomic regions as control. The GWAS SNPs associated all diseases/traits (**d**), were separated into two subsets, associated with blood-related diseases/traits (**e**) or other traits (**f**). The number of enhancers overlap with SNPs in each group was labelled on each bar. $P$ values were calculated using Fisher's exact test. *$P < 0.05$; **$P < 0.01$; ***$P < 0.001$, n.s. not significant. **g–i** Enrichment of GWAS SNPs in hub and non-hub enhancers, which were defined based on chromatin interactions after filtering a specific subtype of chromatin interactions, enhancer-CTCF (**g**), enhancer-enhancer (**h**) or enhancer-promoter (**i**). The fold-change between hub and non-hub enhancers were labelled

from the ICGC database[31]. Hub-enhancer-associated CTCF sites display comparable rates of somatic mutations as TAD boundaries-associated CTCF sites, which are significantly higher than non-hub enhancer-associated CTCF sites ($P = 9.0E-3$ in K562, $P = 2.3E-2$ in GM12878, Fig. 3h and Supplementary Fig. 4h). Our results suggest that genetic alterations of hub enhancer-associated CTCF sites may confer similar consequences as perturbations of TAD boundary-associated CTCF sites, such as activation of proto-oncogenes[28,29].

To get a more comprehensive view of hub enhancers in regulating gene expression, we further identified the enhancer-promoter mappings in K562 cells based on chromatin interactions within TADs. We found that a hub enhancer on average interacts with 1.5 target gene promoters, which is significantly higher than a non-hub enhancer (mean = 1.0, $P = 2.7E-4$, Student's $t$-test), while the enhancers within SEs interact with more target gene promoters than regular enhancers (Supplemen-

tary Fig. 5a). By incorporating transcriptomic data, we found that genes targeted by SEs show higher expression level and cell-type expression specificity than RE-associated genes (Supplementary Fig. 5b,c), which is consistent with previous studies[3]. Importantly, we also observed that genes targeted by hierarchical SEs show higher expression level and cell-type expression specificity than non-hierarchical SEs-associated genes. However, in this analysis we cannot distinguish the roles of hub and non-hub enhancers since they usually target the same set of genes.

Taken together, our results support a model that hub enhancers have two molecularly and functionally related roles in SE hierarchy (Fig. 3i). Hub enhancers act as 'conventional' enhancers to activate gene expression through the recruitment of lineage-specifying transcriptional regulators and coactivators. In addition, they act as 'organizational' hubs to mediate and/or facilitate long-range chromatin interactions through the recruitment of cohesin and CTCF complexes.

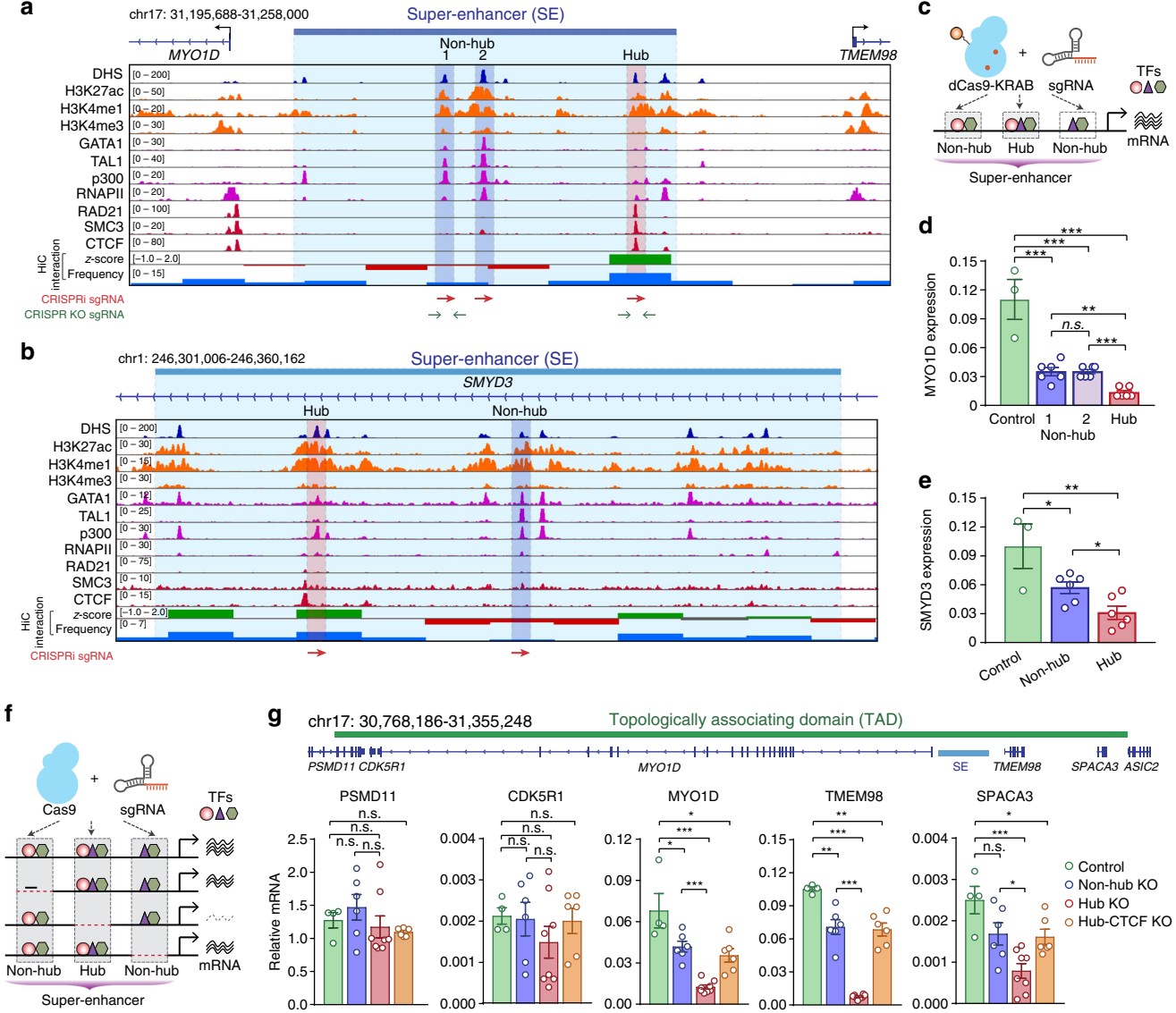

**Fig. 5** In situ genome editing reveals distinct requirement of hub vs non-hub enhancers in SE function. **a** Chromatin signatures and TF occupancy at the MYO1D SE locus in K562 cells are shown. The identified hub and non-hub enhancers are depicted by red (hub) and blue (non-hub) lines, respectively. The Hi-C chromatin interaction z-score and frequency at 5 kb resolution is shown at the bottom. The positions of sgRNAs used for CRISPRi or CRISPR-Cas9-mediated knockout analyses are shown as arrowheads. **b** Chromatin signatures and TF occupancy at the SMYD3 SE locus in K562 cells are shown. **c** Schematic of CRISPRi-mediated repression of hub or non-hub enhancers. **d**, **e** Expression of MYO1D and SMYD3 mRNA in untreated (control), CRISPRi-mediated repression of hub or non-hub enhancers. The mRNA expression levels related to GAPDH determined by qRT-PCR are shown. Each colored circle represents an independent biological replicate experiment. Results are means ± s.e.m. of 3 or 6 independent experiments. *P* values were calculated by two-sided Student's *t*-test. *$P < 0.05$, **$P < 0.01$, ***$P < 0.001$, n.s. not significant. **f** Schematic of CRISPR-Cas9-mediated knockout of hub or non-hub enhancers. **g** Expression of all genes within the SE-containing TAD domain in unmodified (control), CRISPR-Cas9-mediated knockout of hub, non-hub enhancers or the CTCF binding site within the hub enhancer. The mRNA expression levels relative to GAPDH are shown. Each colored circle represents an independent single-cell-derived homozygous enhancer knockout clone. A schematic of the SE-containing TAD domain and associated genes are shown on the top. Results are means ± s.e.m. of at least 4 independent experiments. *P* values were calculated by a two-sided Student's *t*-test. *$P < 0.05$, **$P < 0.01$, ***$P < 0.001$, n.s. not significant

**Hub enhancers are enriched for disease-associated variants.** Genetic variations colocalized with regulatory genomic elements often associate with variation in expression of the linked target genes. As such, expression quantitative trait loci (eQTL) enrichment analysis serves as an objective and quantitative metric to evaluate regulatory potential. We compared the frequencies of eQTLs that are significantly associated with gene expression from the GTEx eQTL database[32] with hub, non-hub and regular enhancers (Fig. 4a and Supplementary Fig. 6a). To measure the enrichment of eQTLs, we defined an enrichment score for each

group of enhancers as the fold enrichment of eQTLs within the group relative to genome background (see Methods). We found the enrichment scores for SEs in K562 and GM12878 (2.1-fold and 2.5-fold) are significantly higher than those for REs (1.5-fold and 1.7-fold) ($P = 1.1E-33$ in K562, $P = 2.2E-59$ in GM12878, Fisher's exact test, Supplementary Fig. 7a). Furthermore, within SEs, hub enhancers are more enriched with eQTLs (2.1-fold and 2.9-fold) compared to non-hub enhancers (2.0-fold and 2.5-fold) ($P = 3.7E-1$ in K562; and $P = 1.5E-2$ in GM12878, Fisher's exact test, Fig. 4a and Supplementary Fig. 6a). A more refined

analysis indicated that, the subset of blood-cell-associated eQTLs are more significantly enriched (5.4-fold and 6.2-fold) than other eQTLs (2.1-fold and 2.9-fold) ($P = 1.3\text{E}{-}2$ in K562, $P = 1.2\text{E}{-}2$ in GM12878, Fisher's exact test, Fig. 4b, c and Supplementary Fig. 6b, c).

To gain additional insights into the function of hub enhancers, we next compared the enhancer groups with genome-wide association study (GWAS)-identified disease-associated genetic variants. Specifically, we analyzed the enrichment of single-nucleotide polymorphisms (SNPs) linked to diverse phenotypic traits and diseases in the GWAS catalog[33]. In a manner similar to the eQTL analysis, the GWAS SNP enrichment scores for SEs in K562 and GM12878 (2.7-fold and 4.8-fold) are significantly higher than those in REs (1.6-fold and 1.9-fold; $P = 3.6\text{E}{-}4$ in K562, $P = 3.0\text{E}{-}15$ in GM12878, Fisher's exact test, Supplementary Fig. 7a). The enrichment of GWAS SNPs at SEs is consistent with previous studies that SEs are enriched with disease-associated variants[3,34]. Importantly, within SEs, hub enhancers display higher enrichment (6.4-fold and 6.8-fold) than non-hub enhancers (2.5-fold and 4.5-fold) ($P = 2.1\text{E}{-}2$ in K562, $P = 1.3\text{E}{-}1$ in GM12878, Fisher's exact test, Fig. 4d and Supplementary Fig. 6d). Furthermore, hub enhancers in K562 cells display much higher enrichment of GWAS SNPs associated with blood traits (22.4-fold) than other SNPs (4.7-fold), even though the difference is not statistically significant due to the small sample size ($P = 5.7\text{E}{-}2$, Fisher's exact test, Fig. 4e, f). We further refined the analysis by leaving out various subtypes of chromatin interactions to evaluate their contributions. We found that the enhancer-CTCF chromatin interactions are most important; leaving them out leads to a decrease of enrichment score from 2.6-fold to 1.4-fold, while other types of interactions have lesser impact (Fig. 4g–i). Of note, in these analyses (Fig. 4 and Supplementary Fig. 6, 7), the hub enhancers in both K562 and GM12878 cells consistently display the highest enrichment of eQTLs and GWAS SNPs compared to non-hub and regular enhancers, although the difference in some comparisons are not statistically significant due to the low numbers of eQTLs/SNPs.

Taken together, our studies demonstrate that hub enhancers are more enriched with genetic variants associated with diseases and cell-type-specific gene expression than other elements within SEs, suggesting they may play a more important role in developmental control and mediating disease risks.

**The model is robust and broadly applicable**. We thoroughly evaluated the robustness of our findings using three complementary criteria. First, to test the robustness of these results with respect to the specific choice of $H$-score threshold, we repeated our analysis by using various thresholds of $H$-score (1.25 and 1.75). The resulting patterns (Supplementary Fig. 7b, c) are similar to our original analysis using the threshold of $H$-score = 1.5 (Fig. 4a, d and Supplementary Fig. 6a, d), suggesting that the properties of hub enhancers are not dependent on the specific threshold of $H$-score.

Since both GM12878 and K562 are hematopoietic in orgin, we tested whether similar patterns can be observed for other cell lineages. Therefore, we analyzed three non-hematopoietic cell lines, including IMR90 (Human Fetal Lung Fibroblasts), HMEC (Human Mammary Epithelial Primary Cell) and HUVEC (Human Umbilical Vein Endothelial Primary Cell) cells. Despite the lower resolution of Hi-C data in these cell lines, we observed a similar trend, that is, hub enhancers are more strongly associated with CTCF binding, GWAS SNPs, and eQTLs than non-hub enhancers (Supplementary Fig. 8).

Finally, we evaluated the robustness of our results with respect to differences in experimental assays by comparing with ChIA-

PET data analysis[19]. To account for the differences in experimental assays, we made a minor modification in defining hierarchical SEs and hub enhancers (see Methods). In total, we identified 188 and 427 hierarchical SEs in K562 and GM12878, respectively. Among these ChIA-PET based hierarchical SEs, 102 and 227, respectively, overlap with Hi-C based hierarchical SEs ($P < 2.2\text{E-}16$ for both cell lines, Fisher's exact test, Supplementary Fig. 9a). The hub enhancers also significantly overlap ($P < 2.2\text{E-}16$ for both cell lines, Fisher's exact test). Importantly, the ChIA-PET based hub enhancers are also more enriched with disease-associated variants compared to non-hub enhancers (Supplementary Fig. 9b). Taken together, these results provide strong support that our approach is robust and broadly applicable.

**In situ CRISPRi analysis of hub vs non-hub enhancers**. Since the structural organization of chromatin plays a critical role in establishing enhancer activities, we then compared the regulatory potential of hub and non-hub enhancers subjected to genetic perturbation. In prior work, we applied CRISPR-Cas9 based genome-editing to systematically dissect the functional hierarchy of an erythroid-specific SE controlling the *SLC25A37* gene encoding the mitochondrial transporter critical for iron metabolism[6]. Following deletion of each of the three constituent enhancers alone or in combination, we identified a functionally 'dominant' enhancer responsible for the vast majority of enhancer activity[6]. Of note, we found that this 'dominant' enhancer is identified as a hub enhancer and associated with significantly higher chromatin interactions compared to the neighboring non-hub enhancers (Supplementary Fig. 10a). These studies provide initial evidence that hub enhancers may be transcriptionally more potent than non-hub enhancers in gene activation.

To further establish the functional roles of hub enhancers, we performed experimental validation of additional randomly selected hierarchical SEs in K562 cells based on the predictions of our model. We first employed CRISPR interference (CRISPRi) in which the nuclease-dead Cas9 protein (dCas9) is fused to a KRAB (Kruppel-associated box) transcriptional repressor domain[35–37]. Upon co-expression of sequence-specific single guide RNAs (sgRNAs) targeting individual hub or non-hub enhancers in K562 cells, we measured the expression of SE-linked target genes as a readout for the functional requirement for SE activity. We focused on two representative SE clusters located in the proximity of the *MYO1D* and *SMYD3* genes (Supplementary Fig. 10b, c and Fig. 5a, b). Both SEs were predicted to contain hierarchical structure ($H$-score = 2.2 and 1.6 respectively), while their nearest target genes *MYO1D* and *SMYD3* are highly expressed in K562 cells. Moreover, both SEs contain hub and non-hub enhancers within a defined TAD domain (Supplementary Fig. 10b, c). Importantly, whereas CRISPRi-mediated repression of the two non-hub enhancers at the *MYO1D* SE led to modest down-regulation (3.1-fold) of *MYO1D* expression, repression of the hub enhancer significantly decreased *MYO1D* expression by 8.3-fold (Fig. 5c, d). Similarly, CRISPRi-mediated repression of the hub enhancer located in the *SMYD3* SE cluster resulted in more profound downregulation of *SMYD3* expression compared to the non-hub enhancer (Fig. 5e).

**Hub enhancers knockout profoundly decreases gene expression**. To further interrogate the role of hub versus non-hub enhancers in SE structure and function in situ, we employed CRISPR-Cas9-mediated genome engineering to delete individual hub or non-hub enhancers with paired sgRNAs flanking the enhancer elements at the *MYO1D* SE (Fig. 5f and Supplementary Fig. 10d). We then measured the expression of all genes within the same enhancer-containing TAD domain. We observed that 3

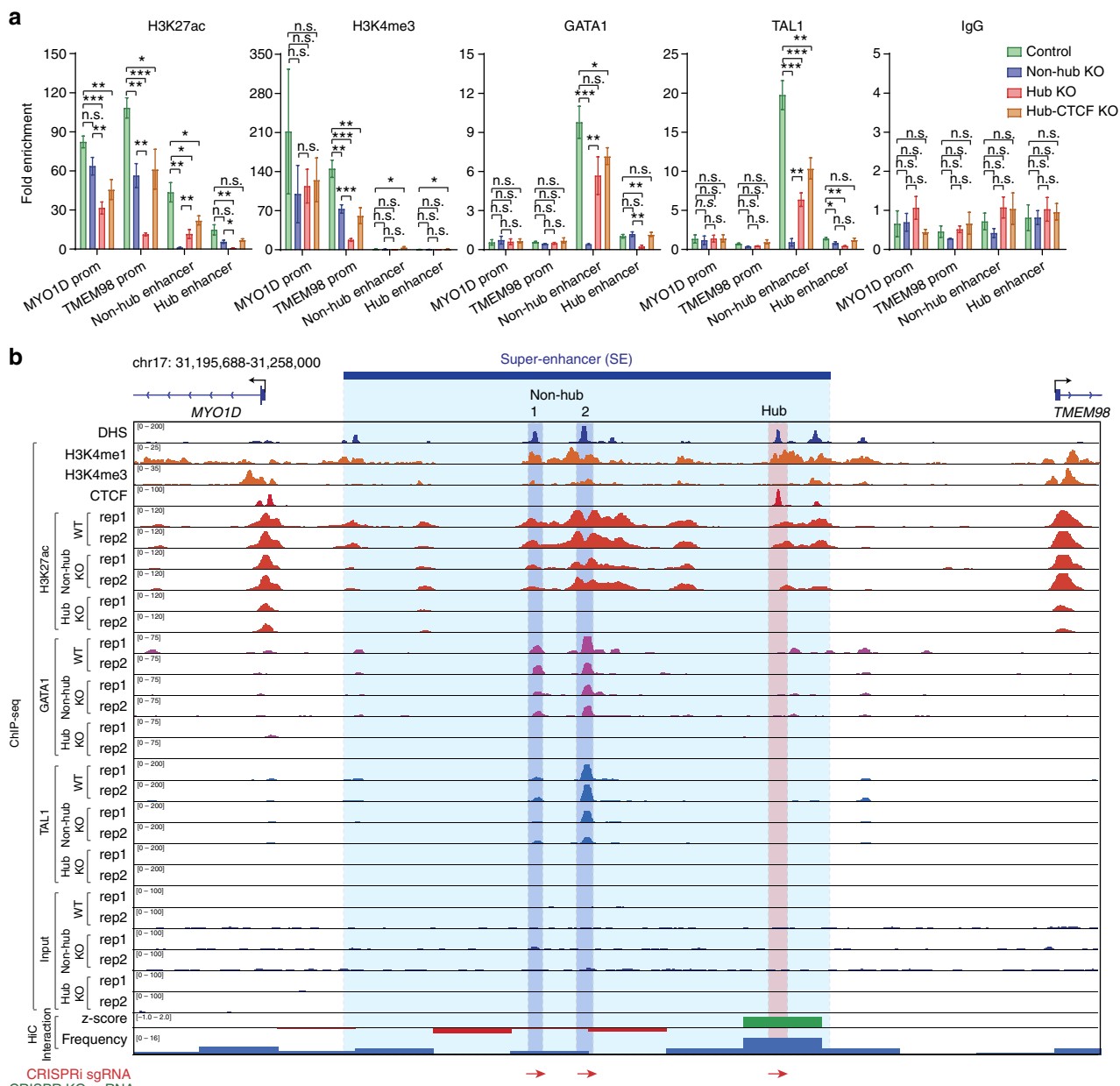

**Fig. 6** Effect on chromatin landscape and TF binding upon KO of the MYO1D hub enhancer. **a** ChIP-qPCR analysis of H3K27ac, H3K4me3, GATA1, TAL1 and IgG (negative control) in unmodified (control), hub, non-hub enhancer or the CTCF binding site within the hub enhancer knockout cells. Primers against MYO1D and TMEM98 promoters, hub and non-hub enhancers, and a negative control genome region (chr2:211,337,339–211,337,429) are used. The results are shown as fold enrichment of the ChIP signals against the negative control region as means ± s.e.m. of four independent experiments. It is important to note that the low or lack of ChIP signals at the non-hub or hub enhancer regions is due to the deletion of the non-hub or hub enhancers containing the primer binding sites. *P* values were calculated by a two-sided Student's t-test. *P < 0.05, **P < 0.01, ***P < 0.001, n.s. not significant. **b** ChIP-seq analysis of H3K27ac, GATA1 and TAL1 in control (WT), hub or non-hub enhancer knockout K562 cells. Browser view of the ChIP-seq intensity from two independent ChIP-seq experiments (rep1 and rep2) is shown. The identified hub and non-hub enhancers are depicted by red (hub) and blue (non-hub) lines, respectively. The Hi-C chromatin interaction z-score and frequency at 5 kb resolution is shown at the bottom (see Methods). The positions of sgRNAs used for CRISPRi or CRISPR-Cas9-mediated knockout analyses are shown as arrowheads

of the 5 genes within the SE-containing TAD domain (*MYO1D*, *TMEM98* and *SPACA3*) displayed significant downregulation in mRNA expression, whereas the other two genes (*PSMD11* and *CDK5R1*) remained unaffected (Fig. 5g and Supplementary Fig. 10b), suggesting that the *MYO1D* SE may regulate only a subset of genes within the same TAD domain. Furthermore, knockout of the hub enhancer resulted in more profound downregulation (5.4, 14.0 and 3.2-fold relative to control; *P* < 0.001) of *MYO1D*, *TMEM98* and *SPACA3* genes compared to the

non-hub enhancers (1.6, 1.5 and 1.5-fold), respectively, consistent with a prominent role of hub enhancers in mediating SE activity.

Our computational analysis showed that CTCF binding is the most distinct feature between hub and non-hub enhancers (Fig. 3). To further establish the functional role of CTCF binding at hub enhancers, we next determined whether deletion of CTCF binding site at hub enhancers influence enhancer activity, target gene expression and/or chromatin landscapes required for transcriptional regulation. To this end, we focused on the

CTCF-occupied hub enhancer at the MYO1D gene cluster (Fig. 5f, g). By CRISPR-Cas9-mediated knockout (KO) using paired sgRNAs, we obtained multiple independent single-cell-derived clones containing homozygous KO of CTCF binding site at the MYO1D hub enhancer (see Methods). Notably, KO of the CTCF binding site at the MYO1D hub enhancer led to significant downregulation of *MYO1D*, *TMEM98* and *SPACA3* genes (Fig. 5g), suggesting that the CTCF binding element at the MYO1D hub enhancer is required for the proper expression of target genes. The relatively modest effect on gene expression upon KO of the CTCF binding site compared to KO of the MYO1D hub enhancer suggests that additional regulatory elements also contribute to the transcriptional activity of the hub enhancer. Taken together, our results provide compelling evidence that the hub enhancer and CTCF binding site at the MYO1D super-enhancer cluster are functionally required for the enhancer activity and expression of target genes.

**Hub enhancers regulate SE local chromatin landscapes**. To determine the effects on the local chromatin landscape and TF binding, we performed ChIP experiments in control (WT), MYO1D hub, non-hub enhancer or the CTCF binding site KO cells (Fig. 6a, b). By quantitative ChIP-qPCR analyses, we observed that KO of the non-hub enhancer had only subtle effects on the enhancer-associated histone mark (H3K27ac) and binding of master TFs (GATA1 and TAL1) at the promoter or enhancer regions of SE-linked *MYO1D* and *TMEM98* genes (Fig. 6a). In contrast, KO of the hub enhancer led to marked downregulation, or near absence, of H3K27ac, H3K4me3, GATA1 and TAL1 binding at neighboring enhancers or promoters. Similarly, KO of the CTCF binding site at the MYO1D hub enhancer also led to downregulation of H3K27ac, GATA1 and TAL1 binding at neighboring enhancer or promoter regions (Fig. 6a), consistent with the downregulation of SE-linked genes (Fig. 5g).

To more comprehensively analyze the effects on chromatin landscape and TF binding, we performed ChIP-seq analysis of H3K27ac, GATA1 and TAL1 in WT, MYO1D hub and non-hub enhancer KO cells (Fig. 6b). By two independent ChIP-seq replicate experiments, we found that KO of the non-hub enhancer (non-hub-1) at the MYO1D SE had no or little effect on the ChIP-seq signals of H3K27ac, GATA1 and TAL1 at the neighboring enhancers (non-hub-2 and hub) or MYO1D/TMEM98 promoters (Fig. 6b). By striking contrast, KO of the hub enhancer led to complete loss of H3K27ac, GATA1 and TAL1 binding at the neighboring enhancers or MYO1D/TMEM98 promoters (Fig. 6b). Furthermore, we observed the changes of H3K27ac, GATA1 and TAL1 at non-hub enhancer (non-hub-1) caused by the hub enhancer KO are more significant than those at hub enhancer caused by the non-hub enhancer KO (Fig. 6b), suggesting that the activity of non-hub enhancers is dependent on the hub enhancer. These results not only validate the ChIP-qPCR analysis but also provide additional molecular evidence that hub enhancers are functionally more potent than neighboring non-hub enhancers in regulating the local chromatin landscape and TF binding, as well as in directing transcriptional activation of SE-linked gene targets (Fig. 5d, g).

Taken together, our in situ genome editing analysis of multiple representative SE clusters provides compelling evidence that at least a subset of SEs are composed of a hierarchical structure containing both hub and non-hub enhancer elements, whereby hub enhancers are functionally indispensable for SE activities.

## Discussion

SE assignment provides a means to identify regulatory regions near important genes that regulate cell fate[5,38–41]. However, it has

remained unclear how SEs function and the extent to which they are distinct from more conventional enhancers. As such, the challenge has been to ascribe functional features uniquely associated with SEs, and account for how the activities of the constituent elements are coordinated for SE function[5]. Here, we have developed a systematic approach to interrogate the structural hierarchy of SE constituent elements based on chromatin interactions. Of note, while a general correlation between chromatin interactions and enhancer activity has been previously established[20–23]. These studies cannot resolve the differences between the constituent elements within a SE.

We observed that only a subset of SEs contains a hierarchical structure, which is consistent with previous findings that SEs are intrinsically heterogeneous, with a large fraction of SEs containing three or fewer constituent elements[5]. Such heterogeneity may provide one explanation for an apparent paradox in the literature[5,42]. For example, recent studies by our group and others provide evidence that SEs may be composed of a hierarchy of enhancer constituents that coordinately regulate gene expression[6,8,9,43,44]. On the other hand, other examples suggest that some SEs may not contain hierarchical structures and the SE constituents contribute additively to gene activation[7,45]. We identified hub enhancers within hierarchical SEs to be associated with an unusually high frequency of long-range chromatin interactions, suggesting that these elements may contribute to the maintenance of SE structure. Moreover, hub enhancers are significantly more enriched with eQTL and GWAS-identified genetic variations, and functionally more potent for gene activation than neighboring non-hub enhancers within the same SEs. Hence, our results support a model in which the structural hierarchy of SEs is predictive of functional hierarchy.

We observed that CTCF binding is highly enriched at hub enhancers compared to other constituent elements. CTCF has an established role in orchestrating genome structure[46]. The prevailing model posits that the primary functions of CTCF are to maintain the boundaries of topological domains and the insulated neighborhoods and to confine the activity of (super-)enhancers and promoters within the boundary[17,47–49]. For example, recent studies have shown that the loss of cohesin or CTCF affects loop domains and transcription[26,50]. However, our results suggest that CTCF play additional, yet important, roles in organizing the structural hierarchy of SEs within TADs. We speculate that the hierarchical organization may be established in a stepwise manner during development through coordinated interactions between CTCF and cell-type specific regulators. Disruption of the hierarchical organization of SE structures may impair SE function and predispose to pathological conditions[28–30]. Consistent with this model, we found that hub-enhancer-associated CTCF binding sites display a significantly higher frequency of somatic mutation than non-hub enhancer-associated CTCF binding sites. Thus, it will be important to investigate chromatin interaction landscapes at both single gene and genomic levels in cancer cells harboring somatic mutations in CTCF binding sites.

At present, Hi-C or ChIA-PET data sets are limited in resolution and available cell types, which presents a significant challenge for further investigation of structural organization within SEs across cell types and conditions. However, the recent development of new technologies, including Hi-ChIP, GAM, capture Hi-C and CAPTURE-3C-seq[23,51–53], promises to enhance the quality and efficiency of data collection for 3D genome structures in various cell types. At the same time, improved methods for functional validation are also being rapidly developed, such as high-resolution CRISPR-Cas9 mutagenesis[43,54]. With anticipated availability of additional chromatin interaction datasets, the computational method we describe here should find wide applications to the systematic investigation of the functional and

structural organization of regulatory elements, including and beyond SEs. Findings from these studies will provide mechanistic insights into the genetic and epigenetic components of human genome in development and disease.

## Methods

**Identification of SEs**. MACS2[55] was used to identify H3K27ac peaks with a threshold $q$-value $= 1.0E{-}5$. H3K27ac peaks were used to define the enhancer boundary, followed by further filtering based on the criteria: (1) excluding H3K27ac peaks that overlapped with ENCODE blacklisted genomic regions[24]; and (2) excluding H3K27ac peaks that were located within ±2 kb region of any RefSeq annotated gene promoter. The remaining H3K27ac peaks were defined as enhancers. Then, SEs were identified by using the ROSE (Rank Ordering of Super-Enhancers) algorithm[2] based on the H3K27ac ChIP-seq signal with the default parameters.

**Analysis of Hi-C data**. High resolution Hi-C data in five human cell types (K562, GM12878, IMR90, HMEC and HUVEC) were obtained from the literature[15]. The statistically significant chromatin interactions in each cell type were detected as previously described[20]. Briefly, the raw interaction matrix was normalized by using the ICE algorithm[56], as implemented in the Hi-Corrector package[57], to remove biases[56,58]. Fit-Hi-C[59] was used to identify statistically significant intra-chromosomal interactions, using the parameter setting '-U = 2000000, -L = 10000' along with the threshold of FDR = 0.01. The interaction frequency for each 5 kb bin was calculated as the number of significant chromatin interactions associated with the bin. The list of TADs in K562 and GM12878 cells were downloaded from the Supplementary Data associated with the publication[15].

**Analysis of chromatin mark distributions**. The sitepro plots for chromatin marks were plotted based on the binned density matrix range from ±5 kb centered by enhancer generated by using the CEAS software[60].

**Analysis of CTCF-related data sets**. Genome-wide CTCF peak locations in 55 cell types, including K562 and GM12878 cells, were downloaded from ENCODE[24]. For each CTCF peak in K562 or GM12878, the cell type consensus score was defined as the percentage of cell types in which the peak was detected.

CTCF motif information, represented as a position weight matrix, was downloaded from the JASPAR database[61]. For each CTCF peak in K562 or GM12878, the corresponding maximum motif-matching score was evaluated by using the HOMER software[62].

The phastCons scores[63] for multiple alignments of 99 vertebrate genomes to the human genome were downloaded from the UCSC Genome Browser. The sitepro plots of conservation score were plotted within ±200 bp centered by CTCF motif sites.

Known somatic mutation loci in cancer were downloaded from International Cancer Genome Consortium (ICGC)[31] Data Portal (release 23). The sitepro plots of mutation frequencies were plotted within ±200 bp centered by CTCF motif sites with a 10 bp smoothing window.

**Enrichment analysis of GWAS SNPs and eQTLs**. The SNPs curated in GWAS Catalog[33] were downloaded through the UCSC Table Browser[64]. The subset of blood-associated GWAS SNPs was selected as those associated with at least one of the following keywords in the 'trait' field: 'Erythrocyte', 'F-cell', 'HbA2', 'Hematocrit', 'Hematological', 'Hematology', 'Hemoglobin', 'Platelet', 'Blood', 'Anemia', 'Sickle cell disease', 'Thalassemia', 'Leukemia', 'Lymphoma', 'Lymphocyte', 'B cell ', 'B-cell', 'Lymphoma', 'Lymphocyte', and 'White blood cell'. Enrichment analysis was carried out as described previously[20]. Briefly, for each group of enhancers, the enrichment score was defined as the fold enrichment relative to genome background. It was calculated as following: $(m/n)/(M/N)$, where $m$ and $M$ represent the number of within-group and genome-wide SNPs respectively, and $n$ and $N$ represent the number of within-group and genome-wide loci respectively. The genome-wide background is estimated from a list of loci generated by randomly shuffling the list of regular enhancers.

Statistically significant eQTL loci in multiple tissues were obtained from the Genotype-Tissue Expression (GTEx) database (accession phs000424.v6.p1)[32]. Blood-associated eQTLs were those identified in the whole blood. eQTLs enrichment analysis was performed similar as those in GWAS SNPs enrichment.

**Analysis of ChIA-PET data set**. CTCF-mediated ChIA-PET data were downloaded from ENCODE[24] (for K562) and the publication website[19] (for GM12878), respectively. The interaction frequency of each 5 kb bin was calculated as the number of chromatin interactions associated the PET clusters located in the bin.

**Analysis of enhancer-promoter mappings based on Hi-C data**. The gene promoters, defined as ±2kb windows centered by RefSeq transcription start site (TSS), were downloaded from the UCSC Genome Browser[64]. The enhancer-gene mappings were identified if enhancer and gene promoter were connected by a

chromatin interaction within a TAD. The normalized gene expression matrix in 57 human cell types was downloaded from Roadmap[65]. The gene expression cell-type specificity in K562 cells was defined as the fold-change of the expression level in K562 comparing with the average expression levels across all cell types.

**Cell culture**. K562 cells were obtained from the American Tissue Collection Center (ATCC). K562 cells were cultured in RPMI1640 medium supplemented with 10% FBS and 1% penicillin–streptomycin.

**CRISPRi of enhancer elements**. The CRISPR-Cas9-mediated interference (CRISPRi) system was used to investigate the function of enhancer elements following published protocol with modifications[35,36]. Briefly, sequence-specific sgRNAs for site-specific interference of genomic targets were designed following described guidelines, and sequences were selected to minimize off-target effect based on publicly available filtering tools (http://crispr.mit.edu/). Oligonucleotides were annealed in the following reaction: 10 µM guide sequence oligo, 10 µM reverse complement oligo, T4 ligation buffer (1×), and 5U of T4 polynucleotide kinase (New England Biolabs) with the cycling parameters of 37 °C for 30 min; 95 °C for 5 min and then ramp down to 25 °C at 5 °C/min. The annealed oligos were cloned into pLV-hU6-sgRNA-hUbC-dCas9-KRAB-T2a-Puro vector (Addgene ID: 71236) using a Golden Gate Assembly strategy including: 100 ng of circular pLV plasmid, 0.2 µM annealed oligos, 2.1 buffer (1×) (New England Biolabs), 20 U of BsmBI restriction enzyme, 0.2 mM ATP, 0.1 mg/ml BSA, and 750 U of T4 DNA ligase (New England Biolabs) with the cycling parameters of 20 cycles of 37 °C for 5 min, 20 °C for 5 min; followed by 80 °C incubation for 20 min. Then K562 cells were transduced with lentivirus to stably express dCas9-KRAB and sgRNA. To produce lentivirus, we plated K562 cells at a density of $3.0 \times 10^6$ per 10 cm plate in high-glucose DMEM supplemented with 10% FBS and 1% penicillin–streptomycin. The next day after seeding, cells were cotransfected with the appropriate dCas9-KRAB lentiviral expression plasmid, psPAX2 and pMD2.G by PEI (Polyethyleneimine). After 8 h, the transfection medium was replaced with 5 ml of fresh medium. Lentivirus was collected 48 h after the first media change. Residual K562 cells were cleared from the lentiviral supernatant by filtration through 0.45 µm cellulose acetate filters. To facilitate transduction, we added the PGE2 (Prostaglandin E2) to the viral media at a concentration of 5 µM. The day after transduction, the medium was changed to remove the virus, and 1 µg/ml puromycin was used to initiate selection for transduced cells. The positive cells were expanded and processed for gene expression analysis.

**CRISPR-Cas9-mediated knockout of enhancer elements**. The CRISPR-Cas9 system was used to introduce deletion mutations of enhancer elements in K562 cells following published protocols[66–68]. Briefly, the annealed oligos were cloned into pSpCas9(BB) (pX458; Addgene ID: 48138) vector using a Golden Gate Assembly strategy. To induce segmental deletions of candidate regulatory DNA regions, four CRISPR-Cas9 constructs were co-transfected into K562 cells by nucleofection using the ECM 830 Square Wave Electroporation System (Harvard Apparatus, Holliston, MA). Each construct was directed to flanking the target genomic regions. To enrich for deletion, the top 1–5% of GFP-positive cells were FACS sorted 48–72 h post-transfection and plated in 96-well plates. Single cell derived clones were isolated and screened for CRISPR-mediated deletion of target genomic sequences. PCR amplicons were subcloned and analyzed by Sanger DNA sequencing to confirm non-homologous end-joining (NHEJ)-mediated repair upon double-strand break (DSB) formation. The positive single-cell-derived clones containing the site-specific deletion of the targeted sequences were expanded for gene expression analysis. To generate small genomic deletions harboring CTCF binding site at the MYO1D hub enhancer without affecting other cis-regulatory elements, we minimized the distance between the paired sgRNAs to 120 bp. The sequences of sgRNAs and genotyping PCR primers are listed in Supplementary Table 1.

**Chromatin immunoprecipitation (ChIP)**. ChIP experiments were performed as described previously[6] with minor modifications. Briefly, $2–5 \times 10^6$ cells were crosslinked with 1% formaldehyde for 5 min at room temperature. Chromatin was sonicated to around 500 bp in RIPA buffer (10 mM Tris-HCl, 1 mM EDTA, 0.1% sodium deoxycholate, 0.1% SDS, 1% Triton X-100, 0.25% sarkosyl, pH 8.0) with 0.3 M NaCl. Sonicated chromatin were incubated with antibody at 4 °C. After overnight incubation, protein A or G Dynabeads (Invitrogen) were added to the ChIP reactions and incubated for four additional hours at 4 °C to collect the immunoprecipitated chromatin. Subsequently, Dynabeads were washed twice with 1 ml of RIPA buffer, twice with 1 ml of RIPA buffer with 0.3 M NaCl, twice with 1 ml of LiCl buffer (10 mM Tris-HCl, 1 mM EDTA, 0.5% sodium deoxycholate, 0.5% NP-40, 250 mM LiCl, pH 8.0), and twice with 1 ml of TE buffer (10 mM Tris-HCl, 1 mM EDTA, pH 8.0). The chromatin was eluted in SDS elution buffer (1% SDS, 10 mM EDTA, 50 mM Tris-HCl, pH 8.0) followed by reverse crosslinking at 65 °C overnight. ChIP DNA was treated with RNaseA (5 µg/ml) and protease K (0.2 mg/ml), and purified using QIAquick Spin Columns (Qiagen). The purified ChIP DNA was quantified by real-time PCR using the iQ SYBR Green Supermix (Bio-Rad). The following antibodies were used: H3K27ac (ab4729, Abcam), H3K4me3 (04–745, Millipore), IgG (12–370, Millipore), GATA1 (ab11852, Abcam), and

TAL1 (sc-12984, Santa Cruz Biotechnology). 1 μg of H3K27ac, H3K4me3 or IgG antibody was used per ChIP experiment, whereas 2 ug of GATA1 or TAL1 antibody was used per ChIP.

**Gene expression measured by qRT-PCR**. Total RNA was isolated using RNeasy Plus Mini Kit (Qiagen) following manufacturer's protocol. qRT-PCR was performed to quantify the target gene expression using the iQ SYBR Green Supermix (Bio-Rad). Relative mRNA expression of target genes was calculated by $2^{-\Delta Ct}$ method, where $\Delta Ct = Ct_{target} - Ct_{GAPDH}$. Primer sequences are listed in Table S1.

**ChIP-seq analysis**. One to 10 ng of ChIP DNA was processed for library generation using NEBNext Ultra II kit following the manufacturer's protocol (New England Biolabs), and sequenced on an Illumina NextSeq500 system using the 75 bp high output sequencing kit. ChIP-seq raw reads were aligned to the hg19 genome assembly using Bowtie2[69] with $k = 1$. The ChIP-seq signals were visualized using Integrative Genomics Viewer (IGV)[70].

**Replicates**. The biological replicates are defined as experiments performed using independently isolated biological samples grown/treated under the same conditions. The technical replicates are defined as experiments performed using the same sample (after all preparatory techniques) and analyzed in multiple times. For the CRISPR-Cas9-mediated KO of hub, non-hub enhancers or the CTCF binding site within the hub enhancer (Fig. 5c–g), independent single cell-derived homozygous KO clones were analyzed, each with two technical replicates. The unmodified control cells were analyzed as two independent biological replicate experiments, each with two technical replicates. For the ChIP-qPCR analysis (Fig. 6a), the results are shown as means ± SEM of two biological replicates, each with two technical replicate measurements. All experimental data points including outliers were included in the data analysis.

**Data availability**. ChIP-seq data of H3K27ac in K562 and GM12878 cells were downloaded from ENCODE[24]. All the data were mapped to the human reference genome version hg19. The 5 kb resolution intra-chromosomal raw interaction matrix in K562 and GM12878 cells were downloaded from a public data set[15]. ChIP-seq data for histone marks (H3K27ac and H3K4me1) and transcription factors/co-activators (GATA1, TAL1, PAX5, EBF1, p300, CTCF, SMC3, and RAD21), DNase-seq in K562 and GM12878 cells were downloaded from ENCODE[24]. All ChIP-seq datasets generated in this study have been deposited in GEO under accession numbers GSE107726. The source code we used to calculate $H$-score and identify hierarchical SEs or hub enhancers is available upon request.

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

## Acknowledgements

We thank Drs. Shiqi Xie and Gary Hon for providing the dCas9-KRAB construct. We thank Dr. Alan Cantor and members of the Yuan Lab for helpful discussions. This work was supported by NIH/NIDDK grants K01DK093543, R03DK101665 and R01DK111430, by a Cancer Prevention and Research Institute of Texas (CPRIT) New Investigator award (RR140025), by the American Cancer Society (IRG-02-196) award and the Harold C. Simmons Comprehensive Cancer Center at UT Southwestern, and by an American Society of Hematology Scholar Award (to J.X.). G.C.Y.'s research was supported by the NIH grants R01HL119099 and R01HG009663.

## Author contributions

J.H., K.L., J.X. and G.-C.Y. conceived and designed the experiments. J.H. and Y.Z. performed bioinformatic analyses. K.L. and X.L. performed experimental validation. J.H., J.X., G.-C.Y., K.L., W.C. and S.H.O. wrote the manuscript. J.X. and G.-C.Y. supervised the project.

## Additional information

**Competing interests:** The authors declare no competing interests.

