## [Peer Review File · Nature Communications]

Reviewers' comments:

Reviewer #1 (Remarks to the Author):

In this manuscript, Huang et al. developed a systematic approach to study the structural hierarchy of super-enhancer (SE) constituent elements based on chromatin interactions. The authors first characterized the intrinsic properties of SEs in the mammalian genome into hierarchical and non-hierarchical enhancers, subclassifying hierarchical enhancers into hub and non-hub. They found that hub enhancers are the major constituent responsible for SE functional and structural organization, as cohesion, CTCF binding sites, cell-type-specific gene expression, and disease-correlated genetic variants are significantly associated with hub enhancers. To further establish the functional roles of hub enhancers, the authors employed in situ CRISPR which both confirmed the role of hub enhancers in SE-associated gene activation and establishment of the local chromatin landscapes. Overall, their work not only revealed how SEs function and the extent to which they are distinct from conventional enhancers, but also proposed an intriguing model in which the structural hierarchy of SEs is predictive of function. The findings presented in this study overcome the technical challenge of systematically characterizing SE signatures on a whole genome scale. The authors pose an intriguing hypothesis where hub enhancers have two distinct, yet related roles; they not only act as traditional enhancers to promote gene activation but also have a distinct role in mediating/establishing long range chromatin interactions and/or 3D configuration. Overall, I feel that this interesting and high quality work is suitable for publication in Nature Communications. I have a few comments as listed below. The authors should address, or at least discuss, these questions prior to publication.

Comment 1: The authors compared the spatial patterns of histone marks, transcription factor binding profiles, as well as cohesin and CTCF occupancy among three enhancer groups : hub, non-hub and regular enhancers (REs). The analysis and signalplots raised a few questions:

1. What if the authors add the non-hierarchical and REs together as another control group? This may better address whether different hierarchical organizations of SEs have impact on spatial distributions for histone marks and TFs.
2. While following the author's analysis pipeline for defining hierarchical and non-hierarchical SEs, I wondered what was the magnitude of similarities and differences between non-hub and non-hierarchical enhancers. There is the possibility that : in one hierarchical SE, only one 5kb-bin is higher than threshold , thereby the remaining non-hub enhancers are the same as other non-hierarchical SEs in terms of H-score. However, they should actually be different, as the hub SE would change the modification distributions, TF binding, and 3D structure around it. This is also why I have suggested using non-hierarchical and REs as a control group.
3. In the TF, cohesin and CTCF binding signalplots, there is a clear dip in the center of the hub enhancer regions, however the authors did not provide an explanation for this. The authors defined hub enhancers which overlapped with the bins of the highest H-score, however they did not give us any information about the length of these hub enhancers, which may have impacted the results. For example, if the hub enhancers in the signalplots are very short in length, there is possibility that the peaks appear on the boundaries of those regions, thereby explaining why there were dips in the figure. For this reason, it would be better if the authors dig deeper into these unusual parts of the figures.

Comment 2: The data source in which the authors applied the pipeline to dissect the SE hierarchy also raised a few questions.

1. The authors only employed 2 human cell lines K562 (erythroleukemia cells) and GM12878 (B-lymphoblastoid cells) because of the limited availability of public, high-resolution Hi-C and ChIP-seq data. Even they are two distinct cell lines, they arise from the same lineage, which inevitably creates a bias. To further validate the hypothesis and extend application of the authors' analysis

approach, they should include data from additional cell lines and tissue samples to achieve more generalizable findings.

2. According to the numbers in line 85 and line 96 (in ms), we find that a similar number of SEs were defined in two different cell lines (843 and 834 SEs in K562 and GM12878 cells), while the hierarchical SEs numbers differ greatly (198 and 286 hierarchical SEs in K562 and GM12878 cells). Is it possible for distinct cell types to have similar SEs but different hierarchical SEs? If this finding is confirmed in other cell lines, it may suggest that hierarchical SEs do correlate more closely with cell-identity and cell-type-specific biological processes.

Minor comments:

1. The authors wrote in line 93 (in ms), 'hub enhancers display a higher frequency of chromatin interactions than non hub enhancers'. However, according to the definition of hub and non-hub, they were already filtered with different H-score (standard chromatin interactions score), which means of course higher frequency in hub ones as the authors made this as such.

2. Is it possible that with different H-score thresholds, the results would vary dramatically? The authors should offer more supporting information on how and why 1.5 (95th percentile of z-scores) was used as the cutoff.

3. Because the authors have the Hi-C data, they should identify the gene promoters which are linked with enhancer interactions. It may also be interesting for the authors to incorporate transcriptome data into their analyses, in order to provide a more cohesive and multilayer view.

Reviewer #2 (Remarks to the Author):

This interesting manuscript addresses the correlation of the hub enhancer within a super-enhancer to chromatin structure by CTCF and cohesin, key components of TAD. There are still relatively few studies on the mechanisms of interaction of cell-specific super-enhancers and their coordination by higher order chromatin structure, hence this work has some novelty. However, though the experiments performed are robust and carefully executed, the technology and the targets used are insufficient to reveal the key functional and structural relationship of a hub-enhancer and CTCF. For example, these experiments used the GEO data set for ChIP-seq, Hi-C and ChIA-PET to make a hypothesis. However, key experiments to confirm the authors' hypothesis have been conducted using ChIP-experiments in mutant cells using CRIPR/Cas9 technologies. The ChIP-experiments conducted by the researchers of this paper inherently provide a very low resolution. The scientific standard is ChIP-seq and it is the only way to unequivocally determine the presence and absence of protein binding "peaks".

The authors propose that CTCF, SMC3 and RAD21 ChIP-seq signals are enriched in the hub enhancer and this could be tested. It would have been scientifically valid to check the chromatin features and chromatin interactions in the respective mutants of hub enhancer and specific CTCF binding sites using appropriate techniques, such as ChIP-seq and Hi-C. Several groups, including the labs of Young, Reinberg and others, have elegantly shown that this is the preferred route. Notably, studies by Drs. Young and Merckenschlager have demonstrated that CTCF sites have biological functions at the TAD boundaries. These deletions of specific sites are the reliable way to validate functional significance.

Further specific comments

1. There are several papers (PMID: 28778681, 26527277, 25303531, and so on) that have studied CTCFs demarcate super-enhancers, and confine the activity of enhancer and promoter within the boundary, and affect the regulation of genes therein. However, the authors addressed that hub enhancers have the motif and binding activity of CTCF inside, and mediate and/or facilitate long-range chromatin interactions through the recruitment of CTCF and cohesin. To prove the hypothesis, the authors need appropriate references and experiments.

2. The authors stated: "Hub enhancers are distinctly associated with cohesin and CTCF binding sites" in the abstract, "Our findings also identified a critical role for CTCF in organizing the structural (and hence functional) hierarchy of SEs" in the introduction, and "We identified those hub enhancers within hierarchical SEs to be associated with an unusually high frequency of long-range chromatin interactions, suggesting that these elements may contribute to the maintenance of SE structure" in the discussion. Also, the authors address that CTCF plays important roles in organizing the structural hierarchy of SEs. However, there is no evidence in this paper to confirm this conclusion. To validate the conclusion, the authors need to show the disruption of interaction between CTCF and enhancers in the mutants with a CTCF binding site deletion and Hub enhancer deletion.

3. The authors cited the paper (PMID: 25547603) to say, "Despite the proposed prominent roles, the structural and functional differences between SEs and REs remain poorly understood" in the Introduction. However, the paper was published in 2015 and more relevant recent papers, for example PMID: 27153539, 28359147, 28340338, 28740262, and so on, to support the authors' opinion are not mentioned in the manuscript.

4. The authors divided each SE into 5kb bins in Fig. 1b. However, the distances between individual enhancers within a super-enhancer are diverse, thereby it does not make sense to divide enhancers within each SE into 5kb. To persuade the concept, a logical explanation would be needed. Also, the authors describe a hub-enhancer has a higher H-score of chromatin interactions. If it is possible, it could be better to analyze the interaction data into three groups, interaction with enhancer, promoter and CTCF.

5. In Fig. 2, the authors address that hub and non-hub enhancers show different quantitative occupancy for active histone markers and TFs. Also, the authors stated hub enhancers are slightly more enriched for H3K27ac and DNase I hypersensitivity but there is no significant difference in H3K4me1. However, considering experimental variation and the distance covering enhancer activity, they seem to be similar and insignificant, and are overlapped in the enhancer center.

6. In Fig. 2, the authors address that transcription factors, such as GATA1, TAL1 and p300, are enriched in hub enhancers compared to non-hub enhancer. However, in Fig. 5 and Supplementary Fig. 8, the hub enhancers in the target gene locus show different features. Especially, in Fig. 5a and Supplementary Fig. 8a, the hub enhancers have little to no activity compared to non-hub enhancers based on H3K27ac. This evidence does not agree with the suggestion of this study.

7. The authors address that CTCF and cohesin, such as SMC3 and RAD21, are abundant at hub enhancers compared to non-hub enhancers in Fig. 3a-c. However, this tendency is not shown in Fig. 5b and Supplementary Fig 8, unlike Fig. 5a and 3a-c. Also, the authors describe "a critical role of CTCF and cohesin in mediating chromatin interactions associated with hub enhancers". However, these ChIP-seq data cannot explain whether the CTCF and cohesin are critical factors in hub-enhancers. Therefore, the authors need to examine whether the deletion of CTCF binding sites within the hub-enhancer fails to activate hub-enhancer and the formation of hierarchical SE using ChIP-seq and Hi-C.

8. In Supplementary Fig. 8c, a TAD is located within the SMYD3 gene without any ChIP-seq peaks for CTCF, SMC3, and RAD21. It does not make sense to identify this region as a TAD.

9. In Fig. 4 and Supplementary Fig. 5 and 6, the authors stated hub enhancers are more enriched with genetic variants based on eQTL. However, there is no significant difference or p-value between hub and non-hub, especially in Fig. 4a and c.

10. In the materials and methods section, the authors missed the explanation for gene expression experiments. Also, in Fig. 5d, e and g, I'm unsure what the numbers to show the gene expression

level mean.

11. In Fig. 5h, the authors show the binding activity of histone markers and master TFs using ChIP-qPCR analysis. However, these data cannot explain the binding pattern in the whole SE. Therefore, the authors need to examine whether the deletion of the hub enhancer causes failure in the recruitment of histone markers and TFs using C.

Dear Editors and Reviewers:

We thank the reviewers for their thoughtful and insightful comments on our manuscript entitled “Dissecting super-enhancer hierarchy based on chromatin interactions”. The reviewers have commented that our study “proposed an intriguing model in which the structural hierarchy of SEs is predictive of function”, where “there are still relatively few studies”, “this interesting and high quality work” (Reviewer #1), “the experiments performed are robust and carefully executed” (Reviewer #2). However, the reviewers also raised some important concerns which we have adequately addressed in this revision by additional computational and experimental analyses.

The following is our point-by-point response to the reviewers’ comments.

Reviewers' comments:

Reviewer #1 (Remarks to the Author):

In this manuscript, Huang et al. developed a systematic approach to study the structural hierarchy of super-enhancer (SE) constituent elements based on chromatin interactions. The authors first characterized the intrinsic properties of SEs in the mammalian genome into hierarchical and non-hierarchical enhancers, subclassifying hierarchical enhancers into hub and non-hub. They found that hub enhancers are the major constituent responsible for SE functional and structural organization, as cohesion, CTCF binding sites, cell-type-specific gene expression, and disease-correlated genetic variants are significantly associated with hub enhancers. To further establish the functional roles of hub enhancers, the authors employed in situ CRISPR which both confirmed the role of hub enhancers in SE-associated gene activation and establishment of the local chromatin landscapes. Overall, their work not only revealed how SEs function and the extent to which they are distinct from conventional enhancers, but also proposed an intriguing model in which the structural hierarchy of SEs is predictive of function. The findings presented in this study overcome the technical challenge of systematically characterizing SE signatures on a whole genome scale. The authors pose an intriguing hypothesis where hub enhancers have two distinct, yet related roles; they not only act as traditional enhancers to promote gene activation but also have a distinct role in mediating/establishing long range chromatin interactions and/or 3D configuration. Overall, I feel that this interesting and high quality work is suitable for publication in Nature Communications. I have a few comments as listed below. The authors should address, or at least discuss, these questions prior to publication.

We thank the reviewer for his/her good summary of this work and valuable suggestions.

Comment 1: The authors compared the spatial patterns of histone marks, transcription factor binding profiles, as well as cohesin and CTCF occupancy among three enhancer groups: hub, non-hub and regular enhancers (REs). The analysis and signalplots raised a few questions:

1. What if the authors add the non-hierarchical and REs together as another control group? This may better address whether different hierarchical organizations of SEs have impact on spatial distributions for histone marks and TFs.

We have now added non-hierarchical SEs as another control group as suggested. Overall, the TF binding profiles at enhancers within non-hierarchical SEs and non-hub enhancers are highly similar. These findings provide additional support for our model that hub enhancers within hierarchical enhancers play predominant roles in the histone modification distributions, TF binding, and 3D structure of super-enhancers. Accordingly, we have updated Fig. 2a-f, Fig. 3a-c, Supplementary Fig. 3a-f, Supplementary Fig. 4a-c and discussed the results in the revised manuscript.

2. While following the author's analysis pipeline for defining hierarchical and non-hierarchical SEs, I wondered what was the magnitude of similarities and differences between non-hub and non-hierarchical enhancers. There is the possibility that: in one hierarchical SE, only one 5kb-bin is higher than threshold, thereby the remaining non-hub enhancers are the same as other non-hierarchical SEs in terms of H-score. However, they should actually be different, as the hub SE would change the modification distributions, TF binding, and 3D structure around it. This is also why I have suggested using non-hierarchical and REs as a control group.

Refer to Point #1, we have added non-hierarchical SEs as another control group as suggested.

3. In the TF, cohesin and CTCF binding signalplots, there is a clear dip in the center of the hub enhancer regions, however the authors did not provide an explanation for this. The authors defined hub enhancers which overlapped with the bins of the highest H-score, however they did not give us any information about the length of these hub enhancers, which may have impacted the results. For example, if the hub enhancers in the signalplots

are very short in length, there is possibility that the peaks appear on the boundaries of those regions, thereby explaining why there were dips in the figure. For this reason, it would be better if the authors dig deeper into these unusual parts of the figures.

The dip in CTCF signal plots is a result of averaging many regions, where CTCF binding is often adjacent to but not directly overlapping with H3K27ac marked chromatin regions (e.g. nucleosomes). Therefore, the observed dip is expected because the center of CTCF binding site is generally nucleosome-free without significant enrichment of H3K27ac. Similar patterns have been observed for other TFs (Fig. 2d-f, Fig. 3a-c). In addition, similar patterns were observed when we re-plotted TF binding in mESCs using other published datasets (e.g. Whyte *et al. Cell*, 2013, from Dr. Young's lab) as shown below.

We agree that it is of interest to show information regarding the length of hub enhancers. We compared the length distribution of hub and non-hub enhancers and found hub enhancers tend to be broader than non-hub enhancers (Supplementary Fig. 1d in revised figures). Accordingly, we have added Supplementary Fig. 1d and discussed the results in the revised manuscript.

Comment 2: The data source in which the authors applied the pipeline to dissect the SE hierarchy also raised a few questions.

1. The authors only employed 2 human cell lines K562 (erythroleukemia cells) and GM12878 (B-lymphoblastoid cells) because of the limited availability of public, high-resolution Hi-C and ChIP-seq data. Even they are two distinct cell lines, they arise from the same lineage, which inevitably creates a bias. To further validate the hypothesis and extend application of the authors' analysis approach, they should include data from additional cell lines and tissue samples to achieve more generalizable findings.

The reviewer is correct that we chose to focus on two cell lines due to the limited availability of public, high-resolution Hi-C data. We agree that it is of interest to test our hypothesis in other non-hematopoietic lineages. Therefore, we have included in the revised manuscript

additional cell lines in which reasonably high resolution Hi-C datasets are available from Rao *et al.*, *Cell* 2014, including IMR90 (Human Fetal Lung Fibroblasts), HMEC (Human Mammary Epithelial Primary Cell) and HUVEC (Human Umbilical Vein Endothelial Primary Cell). We observed similar results that hub enhancers have stronger associations with CTCF binding, GWAS SNPs, and eQTLs than non-hub enhancers (Supplementary Fig. 8 in revised figures). These new results strongly suggest that our approach is robust and generally applicable to other cell lineages. Accordingly, we have added Supplementary Fig. 8 and discussed the results in the revised manuscript.

2. According to the numbers in line 85 and line 96 (in ms), we find that a similar number of SEs were defined in two different cell lines (843 and 834 SEs in K562 and GM12878 cells), while the hierarchical SEs numbers differ greatly (198 and 286 hierarchical SEs in K562 and GM12878 cells). Is it possible for distinct cell types to have similar SEs but different hierarchical SEs? If this finding is confirmed in other cell lines, it may suggest that hierarchical SEs do correlate more closely with cell-identity and cell-type-specific biological processes.

To systematically evaluate this issue, we compared the range of SEs and hierarchical SEs in five cell lines (K562, GM12878, IMR90, HMEC and HUVEC) and found the numbers of hierarchical SEs range from 198 to 500. We also noticed that the numbers of SEs across different cell types defined in previous study also vary from 117 to 1339 (Hnisz, *et al.*, *Cell*, 2013). Thus, as mentioned by reviewer, variation may correlate with cell-identity and cell-type-specific biological processes. Due to the limited numbers of cell lines and variable quality of available Hi-C data, we can't exclude the possibility that this difference is due to technical variation.

Minor comments:

1. The authors wrote in line 93 (in ms), 'hub enhancers display a higher frequency of chromatin interactions than non-hub enhancers'. However, according to the definition of hub and non-hub, they were already filtered with different H-score (standard chromatin interactions score), which means of course higher frequency in hub ones as the authors made this as such.

We agree and have removed the sentence in the revised manuscript to avoid confusion.

2. Is it possible that with different H-score thresholds, the results would vary dramatically?

The authors should offer more supporting information on how and why 1.5 (95th percentile of z-scores) was used as the cutoff.

This is an important question. We set the threshold H-score at 1.5 because it roughly corresponds to the 95th percentile of z-scores (Fig. 1b). To determine whether different H-score thresholds may influence the properties of hub enhancers defined in our analyses, we tested the robustness of our conclusion with respect to different H-score (1.25 and 1.75). As shown in Supplementary Fig. 6b, c (now Supplementary Fig. 7b, c in revised figures), our analysis shows that the properties of hub enhancers are not sensitive to the specific choice of H-score threshold. Nevertheless, we have provided additional information and justification about the H-score threshold in the revised manuscript. Specifically, we added the sentence “We also found the properties of hub enhancers are not sensitive to the specific choice of H-score threshold, as described in the following sections” in the section “A subset of SEs contains hierarchical structure”. Furthermore, we have added a section “The model is robust and generalizable”, and discussed this important point in the revised manuscript.

3. Because the authors have the Hi-C data, they should identify the gene promoters which are linked with enhancer interactions. It may also be interesting for the authors to incorporate transcriptome data into their analyses, in order to provide a more cohesive and multilayer view. We analyzed the Hi-C data and further identified enhancer-promoter mappings in K562 cells based on chromatin interactions within TADs. We found that a hub enhancer on average interacts with 1.5 target gene promoters, which is significantly higher than a non-hub enhancer (mean = 1.0, $P = 2.7E-4$, Student's t-test), while the enhancers within SEs interact with more target gene promoters than regular enhancers (Supplementary Fig. 5a). By incorporating transcriptomic data, we found that genes targeted by SEs show higher expression level and cell-type expression specificity than RE-associated genes (Supplementary Fig. 5b-c), which is consistent with previous studies (Whyte, et al, Cell 2013; Hnisz, et al., Cell, 2013). Importantly, we also observed that genes targeted by hierarchical SEs show higher expression level and cell-type expression specificity than non-hierarchical SEs-associated genes. However, in this analysis we cannot distinguish the roles of hub and non-hub enhancers, since they usually target the same set of genes. Accordingly, we have added Supplementary Fig. 5 and discussed the results in the revised manuscript.

Reviewer #2 (Remarks to the Author):

This interesting manuscript addresses the correlation of the hub enhancer within a super-enhancer to chromatin structure by CTCF and cohesin, key components of TAD. There are

still relatively few studies on the mechanisms of interaction of cell-specific super-enhancers and their coordination by higher order chromatin structure, hence this work has some novelty. However, though the experiments performed are robust and carefully executed, the technology and the targets used are insufficient to reveal the key functional and structural relationship of a hub-enhancer and CTCF. For example, these experiments used the GEO data set for ChIP-seq, Hi-C and ChIA-PET to make a hypothesis. However, key experiments to confirm the authors' hypothesis have been conducted using ChIP-experiments in mutant cells using CRIPR/Cas9 technologies. The ChIP-experiments conducted by the researchers of this paper inherently provide a very low resolution. The scientific standard is ChIP-seq and it is the

only way to unequivocally determine the presence and absence of protein binding "peaks". The authors propose that CTCF, SMC3 and RAD21 ChIP-seq signals are enriched in the hub enhancer and this could be tested. It would have been scientifically valid to check the chromatin features and chromatin interactions in the respective mutants of hub enhancer and specific CTCF binding sites using appropriate techniques, such as ChIP-seq and Hi-C. Several groups, including the labs of Young, Reinberg and others, have elegantly shown that this is the preferred route. Notably, studies by Drs. Young and Merckenschlager have demonstrated that CTCF sites have biological functions at the TAD boundaries. These deletions of specific sites are the reliable way to validate functional significance.

We thank the reviewer for the constructive feedback. We agree that these suggested experiments would help strengthen the conclusions of our paper. In the past 3 months, we have devoted extraordinary efforts to complete nearly all the analyses and experiments requested by the reviewer. More specifically,

1. To further establish the functional role of CTCF binding at hub enhancers, we employed CRISPR/Cas9-mediated knockout of CTCF binding site within the hub enhancer in the MYO1D SE locus, and measured the changes in target gene expression and chromatin landscape required for transcriptional regulation. We found that KO of the CTCF binding site at the MYO1D hub enhancer led to significant down-regulation of the expression of *MYO1D*, *TMEM98* and *SPACA3* genes (Fig. 5g in revised figures), as well as H3K27ac, H3K4me3 and GATA1/TAL1 binding at neighboring enhancers or promoters (Fig. 6a in revised figures). These new results strongly suggest that the CTCF binding element at the MYO1D hub enhancer is functionally required for the enhancer activity and expression of target genes. Accordingly, we have updated Supplementary Fig. 5g, 6a and discussed the results in the revised manuscript.

2. To more comprehensively analyze the effects on chromatin landscape and TF binding, we performed CHIP-seq analysis of H3K27ac, GATA1 and TAL1 in WT, MYO1D hub and non-hub enhancer KO cells (Fig. 6b). By two independent CHIP-seq replicate experiments, we found that KO of the non-hub enhancer (non-hub-1) at the MYO1D SE had no or little effect on the CHIP-seq signals of H3K27ac and GATA1/TAL1 at the neighboring enhancers (non-hub-2) or MYO1D/TMEM98 promoters (Fig. 6b). By striking contrast, KO of the hub enhancer led to complete loss of H3K27ac and GATA1/TAL1 binding at the neighboring enhancers or MYO1D/TMEM98 promoters (Fig. 6b). Furthermore, we observed the changes of H3K27ac and GATA1/TAL1 at non-hub enhancer (non-hub-1) caused by the hub enhancer KO are more significant than those at hub enhancer caused by the non-hub enhancer KO (non-hub-1) (Fig. 6b), suggesting the activity of non-hub enhancers is dependent on the hub enhancer. These results not only validate the CHIP-qPCR analysis but also provide the additional molecular evidence that hub enhancers are functionally more potent than neighboring non-hub enhancers in regulating local chromatin landscape and TF binding and in directing transcriptional activation of SE-linked gene targets (Fig. 5d,g). Accordingly, we have added Fig. 6b and discussed the results in the revised manuscript.

In the meantime, we are also mindful of the technical challenges of the suggested Hi-C experiment. It remains technically challenging and extremely costly to perform Hi-C experiments with the resolution required for detecting Hub enhancer-associated interaction changes, particularly enhancer-promoter interactions at kilobase resolution. This would require sequencing of 3.5 billion reads as previously estimated (Rao *et al. Cell* 2014). To our knowledge, such high resolution profiling has only been performed by the Aiden group. Furthermore, the Hi-C approach is suitable for genome-scale analysis of chromatin interactions whereas it lacks the resolution needed for the quantitative analysis of single enhancer locus-associated interactions. We also considered an alternative approach, CAPTURE, that we recently developed to unbiasedly analyze single genomic locus-associated long-range DNA interactions (Liu *et al., Cell* 2017, 170:1028-1043). However, the CAPTURE protocol needs to be optimized for the current system which will take significant time and effort. Considering the time constraint, we believe this analysis is beyond the scope of the current study and will pursue this direction in a future study instead. Importantly, our paper does not make any predictions on 3D chromatin structure changes. Therefore, while interesting, this analysis does not affect our conclusions.

Further specific comments

1. There are several papers (PMID: 28778681, 26527277, 25303531, and so on) that have studied CTCFs demarcate super-enhancers, and confine the activity of enhancer and promoter within the boundary, and affect the regulation of genes therein. However, the authors addressed that hub enhancers have the motif and binding activity of CTCF inside, and mediate and/or facilitate long-range chromatin interactions through the recruitment of CTCF and cohesin. To prove the hypothesis, the authors need appropriate references and experiments.

We appreciate the reviewer's thoughtful comments and suggestions. As mentioned by the reviewer, a number of prior papers already studied the role of CTCF in the context of TADs / insulated neighborhoods, whereas our novel finding was that CTCF binding is also highly enriched at hub enhancers. Importantly, we found that more than 90% of CTCF peaks at hub enhancers do not overlap with TAD boundaries (Fig. 3d and Supplementary Fig. 4d).

Therefore, our results suggest a TAD-independent role of CTCF at hub enhancers. As described above, we employed CRISPR/Cas9-mediated knockout of CTCF binding site within the hub enhancer in MYO1D SE locus. As a result, three neighboring genes (*MYO1D*, *TMEM98* and *SPACA3*) showed significant down-regulation. Enhancer marks and TF binding were also reduced at neighboring enhancers or promoters. Therefore, the CTCF binding element at the MYO1D hub enhancer is required. We have also added appropriate references as suggested.

2. The authors stated: "Hub enhancers are distinctly associated with cohesin and CTCF binding sites" in the abstract, "Our findings also identified a critical role for CTCF in organizing the structural (and hence functional) hierarchy of SEs" in the introduction, and "We identified those hub enhancers within hierarchical SEs to be associated with an unusually high frequency of long-range chromatin interactions, suggesting that these elements may contribute to the maintenance of SE structure" in the discussion. Also, the authors address that CTCF plays important roles in organizing the structural hierarchy of SEs. However, there is no evidence in this paper to confirm this conclusion. To validate the conclusion, the authors need to show the disruption of interaction between CTCF and enhancers in the mutants with a CTCF binding site deletion and Hub enhancer deletion.

We agree that CRISPR/Cas9-based CTCF binding site deletion would provide critical evidence to establish the functional role of CTCF in maintaining the structure and function of SEs. As described above (refer to Point #1), we deleted the CTCF binding site within the MYO1D associated hub enhancer, and comparatively analyzed the target gene expression and chromatin landscape. We observed significant down-regulation of target genes and reduced occupancy of enhancer marks and TFs at neighboring enhancers or promoters (Fig.

6a in revised figures). Therefore, the CTCF binding site is functionally required for the maintenance of the enhancer activity and target gene expression.

3. The authors cited the paper (PMID: 25547603) to say, “Despite the proposed prominent roles, the structural and functional differences between SEs and REs remain poorly understood” in the Introduction. However, the paper was published in 2015 and more relevant recent papers, for example PMID: 27153539, 28359147, 28340338, 28740262, and so on, to support the authors’ opinion are not mentioned in the manuscript.

Since the concept of SE was proposed in 2013, numerous papers have been published to study super-enhancers in various model systems. For example, Charlet *et al.* showed that bivalent regions of cytosine methylation and H3K27ac may play an active role at enhancers (Charlet *et al.*, Mol. Cell, 2016). Hnisz *et al.* proposed a phase separation model explains how super-enhancers control transcription (Hnisz *et al.*, Cell, 2017). Boeva *et al.* discovered three types of identity in neuroblastoma cell by analyzing the neuroblastoma super-enhancer landscape (Boeva *et al.*, Nat. Genet., 2016). Ko *et al.* summarized recent findings concerning enhancer function in tissues-specific gene regulation and cancer development (Ko *et al.*, Molecules and Cells, 2017). These papers are properly cited in this revision.

However, as noted by the reviewer, “there are still relatively few studies on the mechanisms of interaction of cell-specific super-enhancers and their coordination by higher order chromatin structure”. In our study, we dissected, for the first time, the compositional organization of SEs based on the patterns of long-range chromatin interactions. We found that a subset of SEs exhibit a hierarchical structure, and hub enhancers within hierarchical SEs play distinct roles in chromatin organization and gene activation. We have discussed our findings in the context of existing literature in the “Discussion” section in the revised manuscript.

4. The authors divided each SE into 5kb bins in Fig. 1b. However, the distances between individual enhancers within a super-enhancer are diverse, thereby it does not make sense to divide enhancers within each SE into 5kb. To persuade the concept, a logical explanation would be needed. Also, the authors describe a hub-enhancer has a higher H-score of chromatin interactions. If it is possible, it could be better to analyze the interaction data into three groups, interaction with enhancer, promoter and CTCF.

We agree that the length of each enhancer element is variable. In fact, the majority of the constitutive enhancer elements are less than 5kb. However, our analysis is constrained by

the fact that the highest resolution of existing Hi-C data is 5kb, therefore it is very difficult to adjust our analysis to match with enhancer size.

We thank the reviewer for the suggestion on more refined analysis of different types of chromatin interactions. As suggested, we refined our analysis in K562 cells by leaving out various subtypes of chromatin interactions to evaluate their contributions. We found that the enhancer-CTCF chromatin interactions are most important; leaving them out leads to a decrease of enrichment score from 2.6-fold to 1.4-fold, while other types of interactions have lesser impact (Fig. 4g-i in revised figures). Accordingly, we have added Fig. 4g-i and discussed the results in the revised manuscript.

5. In Fig. 2, the authors address that hub and non-hub enhancers show different quantitative occupancy for active histone markers and TFs. Also, the authors stated hub enhancers are slightly more enriched for H3K27ac and DNase I hypersensitivity but there is no significant difference in H3K4me1. However, considering experimental variation and the distance covering enhancer activity, they seem to be similar and insignificant, and are overlapped in the enhancer center.

We agree with the reviewer that differences in H3K27ac, DNase I hypersensitivity, and TF signals are also marginal on average, in contrast to the significant differences in CTCF and cohesin binding. We revised the text to be more specific in the revised manuscript as following “The signals for H3K27ac and DNase I hypersensitivity are slightly higher at hub than other types of enhancers (Fig. 2b,c and Supplementary Fig. 3b,c); however, the difference is subtle and we cannot exclude the possibility that it may be caused by experimental variations.”

6. In Fig. 2, the authors address that transcription factors, such as GATA1, TAL1 and p300, are enriched in hub enhancers compared to non-hub enhancer. However, in Fig. 5 and Supplementary Fig. 8, the hub enhancers in the target gene locus show different features. Especially, in Fig. 5a and Supplementary Fig. 8a, the hub enhancers have little to no activity compared to non-hub enhancers based on H3K27ac. This evidence does not agree with the suggestion of this study.

As discussed above, the enrichment of GATA1, TAL1 and p300 at hub enhancer is indeed marginal compared to non-hub enhancers, therefore cannot explain functional differences. The fact that the hub enhancers show even weaker H3K27ac signal compared to non-hub enhancers (Fig. 5a and Supplementary Fig. 8a) supports our model that H3K27ac is not a

reliable determinant for enhancer hierarchy. Accordingly, we have clarified this point in the revised manuscript (refer to Point #5).

7. The authors address that CTCF and cohesin, such as SMC3 and RAD21, are abundant at hub enhancers compared to non-hub enhancers in Fig. 3a-c. However, this tendency is not shown in Fig.5b and Supplementary Fig 8, unlike Fig. 5a and 3a-c. Also, the authors describe “a critical role of CTCF and cohesin in mediating chromatin interactions associated with hub enhancers”. However, these ChIP-seq data cannot explain whether the CTCF and cohesin are critical factors in hub-enhancers. Therefore, the authors need to examine whether the deletion of CTCF binding sites within the hub-enhancer fails to activate hub-enhancer and the formation of hierarchical SE using ChIP-seq and Hi-C.

As described above (refer to Points #1 and #2), we applied a CRISPR/Cas9-mediated assay for targeted deletion of the CTCF binding site within the MYO1D associated hub enhancer, followed by gene expression and ChIP profiling. The results strongly validate our prediction that CTCF binding at the hub enhancers is functionally required for the maintenance of enhancer activity.

In addition, it is important to note that while the tendency of CTCF and cohesin binding at hub enhancers is apparent in genome-scale analysis, the relative distance to the hub enhancers varies significantly. For example, strong CTCF/RAD21/SMC3 binding was observed at the center of the hub enhancer at the MYO1D SE (Fig. 5a), whereas CTCF/SMC3 binding was at the border of the hub enhancer at the SMYD3 SE (Fig. 5b). As in many other studies, the association we observed is only an approximation.

8. In Supplementary Fig. 8c, a TAD is located within the SMYD3 gene without any ChIP-seq peaks for CTCF, SMC3, and RAD21. It does not make sense to identify this region as a TAD.

The TAD list used in our work was obtained from the original paper (Rao *et al.*, Cell 2014).

The TADs are shown in collapsed format, where some TADs can be overlapped.

Alternatively, the TADs can be shown in expanded format, which display the details of TADs (see below). Meanwhile, it is important to note that the TADs defined in (Rao *et al.*, Cell 2014) were based on Hi-C chromatin interactions, and not all TAD boundaries contain significant CTCF binding signals.

9. In Fig. 4 and Supplementary Fig. 5 and 6, the authors stated hub enhancers are more enriched with genetic variants based on eQTL. However, there is no significant difference or p-value between hub and non-hub, especially in Fig. 4a and c.

Our conclusion is based on a consistent trend obtained from multiple analyses, even though in some cases the p-value is not significant due to small sample size. As we discussed in the manuscript, “in these analyses (Fig. 4 and Supplementary Fig. 5,6), the hub enhancers in both K562 and GM12878 consistently display the highest enrichment of eQTLs and GWAS SNPs compared to non-hub and regular enhancers, although the difference in some comparisons are not statistically significant due to the low numbers of eQTLs/SNPs.” In addition, we thoroughly evaluated the robustness of our findings using three complementary criteria, (1) various cell types (K562, GM12878, IMR90, HMEC, HUVEC); (2) varying the parameter (H-score=1.25, 1.5, 1.75); (3) different platforms (ChIA-PET and Hi-C). We found hub enhancers consistently display the highest enrichment of eQTLs and GWAS SNPs compared to non-hub, which strongly indicate that our approach is robust and generally applicable to other model systems. Accordingly, we have added Supplementary Fig. 8 and the section “The model is robust and generalizable” in the revised manuscript.

10. In the materials and methods section, the authors missed the explanation for gene expression experiments. Also, in Fig. 5d, e and g, I’m unsure what the numbers to show the gene expression level mean.

The numbers on Figs. 5d, e and g show the relative mRNA expression level relative to the house-keeping GAPDH gene as determined by quantitative real-time PCR. As suggested, we have added a section “Gene expression measured by qRT-PCR” for the detailed explanation of gene expression experiments in the revised manuscript.

11. In Fig. 5h, the authors show the binding activity of histone markers and master TFs using CHIP-qPCR analysis. However, these data cannot explain the binding pattern in the whole SE. Therefore, the authors need to examine whether the deletion of the hub enhancer causes failure in the recruitment of histone markers and TFs using C.

As described above, to more comprehensively analyze the effects on chromatin landscape and TF binding, we performed ChIP-seq analysis of H3K27ac, GATA1 and TAL1 in WT, MYO1D hub and non-hub enhancer KO cells (Fig. 6b). We found loss of the hub enhancer, but not the non-hub enhancer, has major impact on local chromatin landscape resulting in defective transcriptional activation of super-enhancer-linked genes. These results not only validate the CHIP-qPCR analysis, but also provide the additional molecular evidence that hub enhancers are functionally more potent than neighboring non-hub enhancers in affecting the local chromatin landscape and TF binding and directing transcriptional activation of SE-linked gene targets. Accordingly, we have added Fig. 6b and discussed the results in the revised manuscript.

REVIEWERS' COMMENTS:

Reviewer #2 (Remarks to the Author):

I have read through the revised version of the manuscript. The authors have made a commendable effort to address all the issues raised by the reviewers. The only exception are a few laborious experiments that would take a long time to perform. I tend to agree with the authors that these additional experiments fall out of the main theme of the manuscript. I think this is a very nice piece of work on a very timely and interesting topic. My humble opinion is that the manuscript is appropriate for publication in Nature Communications in its current form.

REVIEWERS' COMMENTS:

Reviewer #2 (Remarks to the Author):

I have read through the revised version of the manuscript. The authors have made a commendable effort to address all the issues raised by the reviewers. The only exception are a few laborious experiments that would take a long time to perform. I tend to agree with the authors that these additional experiments fall out of the main theme of the manuscript. I think this is a very nice piece of work on a very timely and interesting topic. My humble opinion is that the manuscript is appropriate for publication in Nature Communications in its current form.

We thank the reviewers for the positive feedback.